# A CLT for Polynomial GNNs on Community-Based Graphs

**Luciano Vinas**
University of California, Los Angeles
lucianovinas@g.ucla.edu

**Arash A. Amini**
University of California, Los Angeles
aaamini@g.ucla.edu

## Abstract

We consider the empirical distribution of the embeddings of a $k$-layer polynomial GNN on a semi-supervised node classification task and prove a central limit theorem for them. Assuming a community based model for the underlying graph, with growing average degree $\nu_n \to \infty$, we show that the empirical distribution of the centered features, when scaled by $\nu_n^{k-1/2}$ converge in 1-Wasserstein distance to a centered stable mixture of multivariate normal distributions. In addition, the joint empirical distribution of uncentered features and labels when normalized by $\nu_n^k$ approach that of mixture of multivariate normal distributions, with stable means and covariance matrices vanishing as $\nu_n^{-1}$. We explicitly identify the asymptotic means and covariances, showing that the mixture collapses towards a 1-D version as $k$ is increased. Our results provide a precise and nuanced lens on how oversmoothing presents itself in the large graph limit, in the sparse regime. In particular, we show that training with cross-entropy on these embeddings is asymptotically equivalent to training on these nearly collapsed Gaussian mixtures.

## 1 Introduction

Graph Neural Networks (GNNs) are now a key tool for machine learning on graphs. Their success is largely due to the graph convolution operation—also known as message passing or neighbor aggregation—where node features are updated by gathering information from their graph neighbors [12; 15; 23]. This process helps GNNs learn powerful embeddings for tasks like node classification and regression. For graphs with community structure, theory shows that even one aggregation step can improve feature separation between classes by a factor of $\sqrt{\nu_n}$, where $\nu_n$ is the average node degree [3].

Analyzing deep GNNs with multiple aggregation layers ($k > 1$) is important but theoretically difficult. Unlike single aggregations, the resulting features, $\phi^{(k)}$, lose desirable properties such as entry-wise independence. To study these multi-aggregated features, researchers have used techniques like walk-based decompositions, which classify feature contributions by underlying graph walk patterns [6; 18]. For community-based graphs, these methods suggest that while feature cluster centers can separate at a rate of $\nu_n^k$, their standard deviation often grows as $\nu_n^{k-1/2}$.

This paper focuses on Polynomial GNNs (Poly-GNNs). In these models, features $\phi^{(k)} = A^k X$ are created by applying the adjacency matrix $A$, $k$ times to initial node features $X \in \mathbb{R}^{n \times d}$, without any non-linear functions in between. These features $\phi^{(k)}$, when passed through a final linear layer $W$, produce classification scores. Poly-GNNs, despite their simplicity, are not just theoretical ideas. They form the basis of several practical and effective GNNs like APPNP [16], GPR-GNN [7], and models using Chebyshev, Bernstein, and Jacobi polynomials [9; 13; 20]. Such models have achieved strong results, sometimes state-of-the-art, on standard benchmarks [20]; indeed, recent rigorous evaluations demonstrate that simplified aggregators and polynomial baselines remain highly competitive with

39th Conference on Neural Information Processing Systems (NeurIPS 2025).

complex architectures [19]. Therefore, understanding Poly-GNN features offers valuable insights into multi-hop aggregation and the behavior of these common GNN types.

## 1.1 Overview of Our Contributions

In this paper, we undertake a detailed asymptotic analysis of the embeddings generated by $k$-layer Poly-GNNs on community-based graphs as the number of nodes $n$ grows. To stabilize these features, we consider two types of normalized embeddings: the degree-normalized features $\overline{\phi}_i^{(k)} := \phi_i^{(k)}/\nu_n^k$, and the centered and scaled features $\xi_i^{(k)} := \sqrt{\nu_n}(\overline{\phi}_i^{(k)} - \mathbb{E}[\overline{\phi}_i^{(k)}])$. Here, $\nu_n$ is the average degree parameter which we assume tends to infinity. One of our main results is a Central Limit Theorem (CLT) demonstrating that the empirical distribution of $\xi_i^{(k)}$ converges in 1-Wasserstein distance to a centered mixture of multivariate Gaussian distributions.

Building upon this, we further demonstrate that the joint empirical distribution of the uncentered, degree-normalized features $\overline{\phi}_i^{(k)}$ (which are directly used in downstream classifiers) and their corresponding true labels $z_i$ also converges in the 1-Wasserstein distance. Specifically, as $n \to \infty$ and $\nu_n \to \infty$, this distribution approaches that of a random pair $(Z, Y_n)$ where $Z \sim \pi$ (the limiting class proportions) and $Y_n$ conditioned on $Z = \ell$ follows a multivariate Gaussian distribution $N(\mu_\ell, \Sigma_\ell/\nu_n)$. A core contribution of our work is the precise analytical characterization of these limiting class means $\mu_\ell$ and class-conditional covariance matrices $\Sigma_\ell$, expressed in terms of the graph's community structure and initial feature means.

This characterization has profound implications for understanding the training dynamics of GNNs. We prove that training a linear classifier on these Poly-GNN features $\overline{\phi}_i^{(k)}$ using the standard cross-entropy (CE) loss converges to the equivalent optimization problem on this limiting Gaussian mixture. This convergence holds uniformly for the loss function, the gradient path during optimization, and the final learned classifier weights (under mild conditions on weight norms), due to the Lipschitz nature of the CE loss and its gradients with respect to the features. This result provides a strong theoretical basis for the behavior observed when training linear classifiers on GNN embeddings.

Furthermore, our explicit forms for $\mu_\ell$ and $\Sigma_\ell$ reveal a clear and precise mechanism behind the well-known phenomenon of GNN oversmoothing. The mean vectors $\mu_\ell$ involve terms of the form $(J^k M)^T$, while the covariance matrices $\Sigma_\ell$ involve $(J^{k-1} M)^T$, where $J$ is a matrix derived from the graph's inter-community edge probabilities and class proportions, and $M$ represents the initial class feature means. As the GNN depth $k$ increases, the repeated matrix exponentiation $J^k$ (and $J^{k-1}$) acts like a power iteration. This causes both the class means and the dominant eigen-directions of the class covariances to align with a low-dimensional (often 1-D) subspace determined by the leading eigenvector(s) of $J$. Consequently, the feature distributions for different classes, initially potentially well-separated in $d$ dimensions, collapse onto this common, typically 1-D, subspace. This results in a degenerate, poorly separated Gaussian mixture, thereby degrading classification performance. Our analysis, thus, provides a nuanced, quantitative view of oversmoothing in the sparse, large-graph limit.

**Previous Literature**   The related literature for multi-hop aggregation can be broken into three categories: *distributional characterizations*, *oversmoothing phenomenon*, and *performance improvements* on select learning tasks, such as classification or regression.

For distributional characterizations, [22] is closest to our work. In their paper, the author's rely on the setting that $\phi^{(k)}$ is exactly component-wise Gaussian for all $n$. We note this cannot be the case as for $\phi_i^{(1)} = \sum_j A_{ij} X_j$ with Bernoulli $A_{ij}$ and normal $X_j$, $\phi_i^{(1)}$ is a (scaled) mixture distribution.

In the vein of oversmoothing, works [5; 17; 21] show how properly normalized aggregations can still oversmooth in the presence of non-linearities. Oversmoothing in this case can be seen as a consequence of the power iteration collapsing the range onto the Perron eigenvectors of $A$. The works [5; 17] show that non-linearities like ReLU do not help oversmoothing since the ReLU operator is also contractive under the operator norm [11]. In [21] the authors extend these results to also include attention-based non-linearities. Outside of [17], which considers the effects of oversmoothing on a $L = 1$ community graph, all other works assume $A$ is a deterministic graph. Our work differs fundamentally by analyzing a *stochastic* graph model in the large-graph limit. While prior work

often explains oversmoothing via power iteration on a fixed adjacency matrix, showing that feature *means* collapse, our CLT reveals a more powerful mechanism. Building upon our previous work of matrix moment analysis for community-based graphs [18], we prove that the feature *covariance* also collapses onto the same unfavorable, low-dimensional subspace as the means. This provides a much stronger characterization of feature degeneracy.

With respect to improving task performance, works [4; 14] show how multi-hop features can improve downstream learning tasks. Between the two works the generative formulation differs, [4] assumes a $(p, q)$-SBM with mixed mean feature representations while [14] assumes a low rank, latent variable model which yields dense observed graphs. The losses considered by [4; 14] are Lipschitz, highlighting the importance of understanding behavior of multi-hop features under the 1-Wasserstein metric.

## 2 Preliminaries and Model Setup

In this section, we formally define the Polynomial GNN (Poly-GNN) architecture, introduce the normalized features central to our analysis, describe the community-based graph model, state our key assumptions, and briefly define the Wasserstein distance used to quantify distributional convergence.

### 2.1 Poly-GNNs and Feature Definitions

We consider a simple yet powerful class of Graph Neural Networks known as Polynomial GNNs (Poly-GNNs). Given an undirected graph with $n$ nodes, represented by its adjacency matrix $A \in \{0, 1\}^{n \times n}$, and initial node features $X \in \mathbb{R}^{n \times d}$, a $k$-layer Poly-GNN computes node embeddings, or features, $\phi^{(k)} \in \mathbb{R}^{n \times d}$ through $k$ successive aggregations:

$$\phi^{(k)} = A^k X. \tag{1}$$

The $i$-th row of $\phi^{(k)}$, denoted $\phi_i^{(k)} \in \mathbb{R}^d$, represents the embedding for node $i$ after $k$ layers of aggregation.

For our asymptotic analysis, we work with normalized versions of these features. Let $\nu_n$ be the average degree parameter of the graph, which we assume grows with $n$ (see Assumption 1). We define the *degree-normalized features* as:

$$\overline{\phi}_i^{(k)} := \frac{\phi_i^{(k)}}{\nu_n^k}, \quad i = 1, \dots, n. \tag{2}$$

These features $\overline{\phi}_i^{(k)}$ are often the direct input to a downstream classifier. In practice, the unknown parameter $\nu_n$ is not required, as it can be reliably replaced by the observed average degree.

To establish a stable limiting distribution under a Central Limit Theorem, we further define the *centered and scaled features*:

$$\xi_i^{(k)} := \sqrt{\nu_n} \left( \overline{\phi}_i^{(k)} - \mathbb{E}[\overline{\phi}_i^{(k)}] \right), \quad i = 1, \dots, n. \tag{3}$$

The empirical distribution of these features, $\mathbb{P}_n := \frac{1}{n} \sum_{i=1}^n \delta_{\xi_i^{(k)}}$, where $\delta_x$ is a point mass at $x$, will be a primary object of study.

### 2.2 Community-Based Graph Model

We assume the graph and its node features are generated from a community-based model. Let $z = (z_i)_{i=1}^n \in [L]^n$ be a vector of latent node labels, assigning each node $i$ to one of $L$ communities or classes. The graph structure and initial feature distributions are conditional on these labels.

Specifically, we adopt the Contextual Stochastic Block Model (CSBM) [10]. The adjacency matrix $A$ is generated such that edges are conditionally independent given $z$, with probabilities:

$$A_{ij} \sim \text{Bern}(\nu_n B_{z_i z_j}/n) \quad \text{for } i \neq j, \text{ and } A_{ii} = 0, \tag{4}$$

where $B \in [0, 1]^{L \times L}$ is a symmetric matrix of inter-community edge probability scalings. The parameter $\nu_n/n$ represents the average edge density scale.

The initial node features $X_i \in \mathbb{R}^d$ are assumed to be conditionally independent given $z_i$. Their expectations are determined by their class membership:

$$\mathbb{E}[X_i \mid z_i = \ell] = M_{\ell,\cdot}, \tag{5}$$

where $M \in \mathbb{R}^{L \times d}$ is a matrix whose $\ell$-th row, $M_{\ell,\cdot}$, is the mean feature vector for class $\ell$. This can be written compactly as $\mathbb{E}[X] = ZM$, where $Z \in \{0,1\}^{n \times L}$ is the one-hot encoding matrix of the labels $z$, i.e., $Z_{i\ell} = \mathbb{1}\{z_i = \ell\}$.

We define $\pi = (\pi_1, \ldots, \pi_L)^T$ as the vector of limiting class proportions (see Assumption 3). Let $\Pi = \mathrm{diag}(\pi_1, \ldots, \pi_L)$ be the diagonal matrix of these proportions. A key matrix in our analysis is $J \in \mathbb{R}^{L \times L}$, defined as:

$$J = B\Pi. \tag{6}$$

This matrix captures the interplay between inter-community connectivity $B$ and class sizes $\Pi$.

### 2.3 Assumptions

Our theoretical results rely on the following assumptions:

**Assumption 1** (Degree Growth). *The average degree parameter $\nu_n \to \infty$ as $n \to \infty$, and $B_{\ell\ell'} \leq C$ for some constant $C$, implying that the expected degree of any node $i$, $\sum_{j \neq i} \nu_n B_{z_i z_j}/n$, is $O(\nu_n)$.*

**Assumption 2** (Sparse Graph). *The graph is sparse, meaning $\nu_n = o(n)$.*

**Assumption 3** (Cluster Convergence). *For each class $\ell \in [L]$, let $\mathcal{C}_\ell = \{i \in [n] : z_i = \ell\}$ be the set of nodes in class $\ell$. We assume there exist $\pi_\ell > 0$ such that $\pi_\ell - |\mathcal{C}_\ell|/n = o(1)$, and $\sum_{\ell=1}^{L} \pi_\ell = 1$.*

**Assumption 4** (Feature Bounds). *The initial node features $X_i$ are sub-gaussian. Specifically, for any unit vector $u \in \mathbb{R}^d$, $(X_i - \mathbb{E}[X_i])u \sim SG(\sigma^2)$ for some $\sigma^2 > 0$ uniformly for all $i, n$. Furthermore, their expected norms are uniformly bounded: $\limsup_{n \geq 1} \sup_{i \in [n]} \mathbb{E}\|X_i\|_2 \leq x_*$ for some $x_* \geq 0$.*

Of the listed assumptions, Assumptions 1 and 3 are necessary as, without these, a limiting Gaussian distribution cannot be obtained. See Figure 1 for more details on the case of $L = 1$. Assumption 4 is mild and subsumes a large class of feature distributions. Assumption 2 is a simplifying one. Our CLT framework extends to the dense regime ($\nu_n = \Omega(n)$), but the limiting covariance structure becomes more complex. As detailed in Appendix A.1, the variance of the aggregated features decomposes into terms driven by graph randomness ($A$) and initial feature randomness ($X$). In the sparse setting, graph randomness dominates, causing the initial feature covariance to be negligible in the limit (Appendix A.3). In the dense case, this feature-related noise term persists, leading to a different limiting covariance. We focus on the sparse case as it is representative of many large-scale networks.

### 2.4 Wasserstein Distance

To measure the distance between probability distributions, we use the 1-Wasserstein distance, denoted $W_1(\mathbb{P}, \mathbb{Q})$. For two probability measures $\mathbb{P}$ and $\mathbb{Q}$ on $\mathbb{R}^d$, the Kantorovich-Rubinstein duality provides a convenient definition:

$$W_1(\mathbb{P}, \mathbb{Q}) = \sup_{f \in \mathrm{Lip}(1)} \left| \int f d\mathbb{P} - \int f d\mathbb{Q} \right|, \tag{7}$$

where $\mathrm{Lip}(1)$ is the class of all 1-Lipschitz functions $f : \mathbb{R}^d \to \mathbb{R}$, i.e., functions satisfying $|f(x) - f(y)| \leq \|x - y\|_2$ for all $x, y \in \mathbb{R}^d$. We also write $\mathbb{P}f = \int f d\mathbb{P}$ for the expectation of $f$ under $\mathbb{P}$. Convergence in $W_1$ implies weak convergence and convergence of first moments. Its connection to Lipschitz functions makes it particularly relevant for analyzing learning algorithms with Lipschitz loss functions.

## 3 Asymptotic Distribution of Poly-GNN Embeddings

In this section, we present our main theoretical results concerning the asymptotic distribution of Poly-GNN embeddings. We establish Central Limit Theorems (CLTs) for both the degree-normalized features $\overline{\phi}_i^{(k)}$ (jointly with their labels) and the centered-and-scaled features $\xi_i^{(k)}$. We then outline the key steps involved in proving these theorems, highlighting the key intermediate lemmas and propositions.

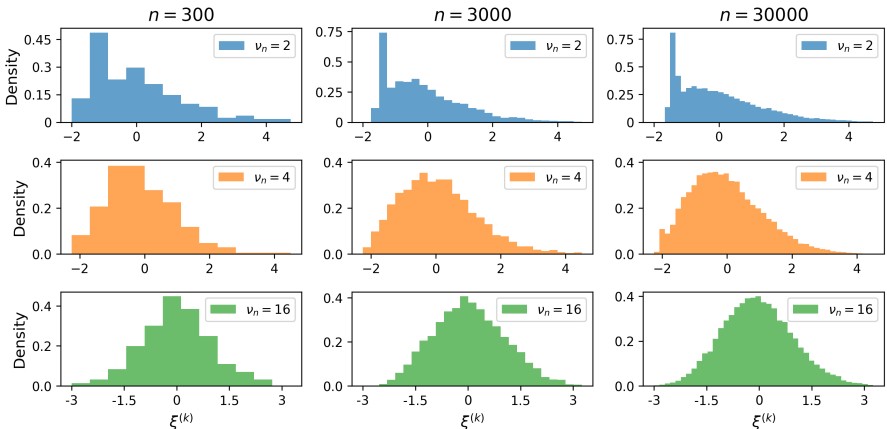

Figure 1: Comparison of the $\xi^{(k)}$ distribution for $k = 3$ across different expected degree Erdős–Réyni graphs. As graph size increases, the overall histogram resolution is improved but this does not qualitatively change the shape of the histogram. That is, growing degree $\nu_n \to \infty$, is a neccesary condition for $\xi^{(k)}$ to be Gaussian.

## 3.1 Main Central Limit Theorems

Our first main result characterizes the joint limiting distribution of the true node labels $z_i$ and the degree-normalized Poly-GNN features $\overline{\phi}_i^{(k)}$. These features are typically used for downstream classification tasks.

**Theorem 1** (CLT for Degree-Normalized Features and Labels). *Let $(A, X)$ be a community-based graph satisfying Assumptions 1–4. Let $\overline{\phi}_i^{(k)} = \phi_i^{(k)}/\nu_n^k$ be the degree-normalized $k$-layer Poly-GNN features. Define the limiting class means $\mu_\ell \in \mathbb{R}^d$ and class-conditional covariance matrices $\Sigma_\ell \in \mathbb{R}^{d \times d}$ as:*

$$\mu_\ell := (J^k M)^T e_\ell, \tag{8}$$

$$\Sigma_\ell := (J^{k-1} M)^T \operatorname{diag}(e_\ell^T J)(J^{k-1} M), \tag{9}$$

*where $e_\ell$ is the $\ell$-th canonical unit vector in $\mathbb{R}^L$, $J = B\Pi$, and $M$ contains the initial class feature means. Let $\widetilde{\mathbb{P}}_n^{joint}$ be the empirical distribution of pairs $(z_i, \overline{\phi}_i^{(k)})$: $\widetilde{\mathbb{P}}_n^{joint} = \frac{1}{n} \sum_{i=1}^n \delta_{(z_i, \overline{\phi}_i^{(k)})}$. Let $\mathbb{G}_n^{joint}$ be the probability distribution of a random pair $(Z, Y_n)$ where $Z \sim \operatorname{Categorical}(\pi_1, \ldots, \pi_L)$ and, conditioned on $Z = \ell$, $Y_n \sim N(\mu_\ell, \Sigma_\ell/\nu_n)$. Then, as $n \to \infty$:*

$$\mathbb{E}\left[W_1\left(\widetilde{\mathbb{P}}_n^{joint}, \mathbb{G}_n^{joint}\right)\right] \to 0. \tag{10}$$

*Furthermore, this convergence holds in the stronger class-conditional sense: for any $R > 0$,*

$$\lim_{n \to \infty} \mathbb{E}\left\{ \sup_{f_1, \ldots, f_L \in \operatorname{Lip}(R)} \left| \frac{1}{n} \sum_{\ell=1}^L \sum_{i \in \mathcal{C}_\ell} f_\ell(\overline{\phi}_i^{(k)}) - \sum_{\ell=1}^L \pi_\ell \mathbb{E}_{Y \sim N(\mu_\ell, \Sigma_\ell/\nu_n)}[f_\ell(Y)] \right| \right\} = 0. \tag{11}$$

Theorem 1 shows that for large $n$ and $\nu_n$, the features $\overline{\phi}_i^{(k)}$ behave as if drawn from a Gaussian mixture where each component $\ell$ is centered at $\mu_\ell$ and has a covariance $\Sigma_\ell/\nu_n$ that vanishes as $\nu_n \to \infty$.

Our second main result provides a CLT for the centered and scaled features $\xi_i^{(k)}$, showing they converge to a stable (non-degenerate variance) Gaussian mixture.

**Theorem 2** (CLT for Centered and Scaled Features). *Under the same conditions as Theorem 1, let $\xi_i^{(k)} = \sqrt{\nu_n}(\overline{\phi}_i^{(k)} - \mathbb{E}[\overline{\phi}_i^{(k)}])$ be the centered and scaled features. Let $\mathbb{P}_n = \frac{1}{n} \sum_{i=1}^n \delta_{\xi_i^{(k)}}$ be their empirical distribution. Let $\mathbb{G}$ be the centered Gaussian mixture distribution:*

$$\mathbb{G} = \sum_{\ell=1}^L \pi_\ell N(0, \Sigma_\ell), \tag{12}$$

*where $\Sigma_\ell$ is defined in Eq. (9). Then, as $n \to \infty$:*

$$\mathbb{E}\left[W_1(\mathbb{P}_n, \mathbb{G})\right] \to 0. \tag{13}$$

*Furthermore, this convergence also holds in the stronger class-conditional sense: for any $R > 0$,*

$$\lim_{n \to \infty} \mathbb{E}\left\{\sup_{f_1,\ldots,f_L \in \mathrm{Lip}(R)} \left| \frac{1}{n} \sum_{\ell=1}^{L} \sum_{i \in \mathcal{C}_\ell} f_\ell(\xi_i^{(k)}) - \sum_{\ell=1}^{L} \pi_\ell \mathbb{E}_{Y \sim N(0,\Sigma_\ell)}[f_\ell(Y)] \right| \right\} = 0. \tag{14}$$

Theorem 2 establishes that after appropriate centering and scaling, the Poly-GNN features converge to a mixture of Gaussians, each component having a non-vanishing covariance $\Sigma_\ell$.

We note that Theorem 1 may be of more interest in practical scenarios, since the uncentered features do not require estimation of the feature mean $\mathbb{E}[\overline{\phi}^{(k)}]$. Furthermore, in settings where Assumption 1 and 3 hold, the average degree $\overline{d}$ becomes a reliable estimate of normalization scale since $\overline{d} \asymp \nu_n$.

## 3.2 Proof Outline and Key Steps

The proofs of Theorems 1 and 2 share a common foundation and proceed in several steps. We outline the general strategy here, focusing on the convergence of $\mathbb{P}_n$ to $\mathbb{G}$ (Theorem 2). The argument for Theorem 1 builds on Theorem 2 with adjustments for the non-zero means and the $\nu_n^{-1}$ scaling in the covariance. The full proofs are provided in Appendix A.

The overall strategy involves two main parts for establishing $\mathbb{E}[W_1(\mathbb{P}_n, \mathbb{G})] \to 0$:

1. Show that the empirical measure $\mathbb{P}_n$ concentrates around its expectation $\overline{\mathbb{P}}_n := \mathbb{E}[\mathbb{P}_n]$, i.e., $\mathbb{E}[W_1(\mathbb{P}_n, \overline{\mathbb{P}}_n)] \to 0$.

2. Show that the expected empirical measure $\overline{\mathbb{P}}_n$ converges to the target Gaussian mixture $\mathbb{G}$ in $W_1$ distance, i.e., $W_1(\overline{\mathbb{P}}_n, \mathbb{G}) \to 0$.

The argument for class-conditional convergence (e.g., Eq. (11)) builds upon this by considering per-class empirical measures and leveraging the convergence of class proportions $|\mathcal{C}_\ell|/n \to \pi_\ell$.

The key technical steps involve analyzing the moments of the features:

**Step 1: Moment Analysis for General Graphs** This step characterizes the behavior of feature moments without yet imposing the full community structure, relying mainly on Assumptions 1, 2, and 4.

- The centered, un-normalized features $\phi_i^{(k)} - \mathbb{E}[\phi_i^{(k)}]$ are decomposed into two terms: $\mathring{\Delta}_i$ (due to graph randomness) and $\mathring{\Lambda}_i$ (due to initial feature randomness):

$$\phi_i^{(k)} - \mathbb{E}[\phi_i^{(k)}] = \mathring{\Delta}_i + \mathring{\Lambda}_i. \tag{15}$$

  Normalizing appropriately, $\xi_i^{(k)} = (\mathring{\Delta}_i + \mathring{\Lambda}_i)/\nu_n^{k-1/2}$.

- The term $\Lambda_i := \mathring{\Lambda}_i/\nu_n^{k-1/2}$ is shown to be asymptotically negligible under our sparsity assumption (see Proposition 2 in Appendix A.3). Thus, $\xi_i^{(k)}$ is asymptotically equivalent to $\Delta_i := \mathring{\Delta}_i/\nu_n^{k-1/2}$ in terms of its contribution to moments (see Lemma 5 in Appendix A.3).

- The moments of $\Delta_{i,\theta} := \langle \Delta_i, \theta \rangle$ for any unit vector $\theta \in \mathbb{R}^d$ are analyzed.
    - Odd moments: $\mathbb{E}[\Delta_{i,\theta}^r] \to 0$ for odd $r$ (this follows from the moment bounds in Proposition 3, specifically the term $\nu_n^{p/2-\lceil p/2 \rceil}$, which is $\nu_n^{-1/2}$ for odd $p = r$).
    - Even moments: $\mathbb{E}[\Delta_{i,\theta}^r] \to (r-1)!! \cdot \widetilde{\sigma}_{i,\theta}^r$ for even $r$, where $\widetilde{\sigma}_{i,\theta}^2 := \|V_i \mathbb{E}[A/\nu_n]^{k-1} \mathbb{E}[X]\theta\|_2^2$ (see Lemma 7 in Appendix A.3).

- The expected normalized mean $\mathbb{E}[\overline{\phi}_i^{(k)}]$ is shown to converge to a limit $\gamma_i = e_i^T (\mathbb{E}[A/\nu_n])^k \mathbb{E}[X]$ (see Lemma 3 in Appendix A.2).

**Step 2: Specialization to Community-Based Graphs** Here, the community structure (Assumptions 3 and the CSBM formulation) is used to refine the limiting moments.

- The limiting mean $\gamma_i$ for a node $i \in \mathcal{C}_\ell$ converges to $\mu_\ell = (J^k M)^T e_\ell$ (as detailed in the proof of Proposition 4 in Appendix A.4, building on Lemma 3).
- The average of the per-node variances $\widetilde{\sigma}_{i,\theta}^2$ over class $\ell$ converges to $\theta^T \Sigma_\ell \theta$, where $\Sigma_\ell$ is defined in Eq. (9) (this is part of the derivation in the proof of Proposition 4).
- Consequently, the $r$-th moment of the $\theta$-projection of $\overline{\mathbb{P}}_n$, $m_r(\overline{\mathbb{P}}_{n,\theta}) = \frac{1}{n} \sum_i \mathbb{E}[\langle \xi_i^{(k)}, \theta \rangle^r]$, converges to $m_r(\mathbb{G}_\theta) = \sum_{\ell=1}^L \pi_\ell \mathbb{E}_{Y \sim N(0, \theta^T \Sigma_\ell \theta)}[Y^r]$ (see Proposition 4 in Appendix A.4).

Since the Gaussian mixture $\mathbb{G}_\theta$ is determined by its moments, this establishes that $\overline{\mathbb{P}}_{n,\theta} \rightsquigarrow \mathbb{G}_\theta$. Uniform integrability of moments (derived from the $\Psi_r$ norm bounds in appendix C, specifically Lemma 9, applied to $\Delta_{i,\theta}$ via Proposition 3) then promotes this weak convergence to $W_1(\overline{\mathbb{P}}_{n,\theta}, \mathbb{G}_\theta) \to 0$. A discretization argument (Proposition 7 from Appendix B) and Proposition 8 (from Appendix D) extend this to $W_1(\overline{\mathbb{P}}_n, \mathbb{G}) \to 0$.

**Step 3: Concentration and Convergence of Empirical Measure.** To show $\mathbb{E}[W_1(\mathbb{P}_n, \overline{\mathbb{P}}_n)] \to 0$, we rely on:

- Control over the variance of empirical moments: $\mathrm{Var}(n^{-1} \sum_{i=1}^n \langle \Delta_{i,\theta} \rangle^r) \lesssim n^{-1}$ (see Lemma 6 in Appendix A.3, which implies similar behavior for $\xi_i^{(k)}$ via Lemma 5). This corresponds to condition (b) of Proposition 8 in Appendix D.
- Tail control for $\langle \xi_i^{(k)}, \theta \rangle$: The features $\langle \xi_i^{(k)}, \theta \rangle$ are shown to be uniformly $\Psi_{r_n}$ sub-Gaussian for a growing $r_n$ (see Lemma 1 in Appendix A.1). This corresponds to condition (a) of Proposition 8.
- Uniform integrability of moments of $\overline{\mathbb{P}}_n$ (the convergence shown in Proposition 4 implies that for any fixed $r$, $\sup_n m_r(\overline{\mathbb{P}}_{n,\theta})$ is finite, which by Proposition 7 implies $\sup_n M_r(\overline{\mathbb{P}}_n)$ is finite, e.g. $M_1(\overline{\mathbb{P}}_n)$ needed for condition (c) of Proposition 8).

These conditions allow the application of Proposition 8 (from Appendix D), which establishes the desired concentration $\mathbb{E}[W_1(\mathbb{P}_n, \overline{\mathbb{P}}_n)] \to 0$. The triangle inequality for $W_1$ then combines these two main parts to yield the final convergence result.

# 4 Implications for Classification and GNN Oversmoothing

The Central Limit Theorems presented in Section 3 not only provide a fundamental understanding of the distributional properties of Poly-GNN embeddings but also have significant practical implications. In this section, we explore two key consequences: first, how our results explain the convergence of linear classifiers trained on these embeddings, and second, how they offer a precise, quantitative mechanism for the GNN oversmoothing phenomenon [5; 17].

## 4.1 Convergence of Linear Classification on Poly-GNN Features

In many node classification tasks, GNN embeddings are fed into a final linear layer (often followed by a softmax activation) that is trained using a cross-entropy (CE) loss. Our results provide a theoretical basis for understanding this training process in the asymptotic limit. We focus on the degree-normalized features $\overline{\phi}_i^{(k)}$, as these are the quantities typically used by the classifier.

Recall from Theorem 1 that the joint empirical distribution of labels $z_i$ and features $\overline{\phi}_i^{(k)}$ converges to that of $(Z, Y_n)$, where $Z \sim \mathrm{Categorical}(\pi)$ and $Y_n \mid Z = \ell \sim N(\mu_\ell, \Sigma_\ell/\nu_n)$. The class means $\mu_\ell$ and covariances $\Sigma_\ell$ are given by Eqs. (8) and (9), respectively.

Consider a linear classifier with weights $W = (w_1, \ldots, w_L)^T \in \mathbb{R}^{L \times d}$ and biases $b = (b_1, \ldots, b_L)^T \in \mathbb{R}^L$. The empirical cross-entropy loss for a dataset of $n$ nodes is:

$$\mathcal{L}_{\mathrm{emp}}(W, b) := -\frac{1}{n} \sum_{i=1}^n \sum_{\ell=1}^L \mathbb{1}\{z_i = \ell\} \log \frac{\exp(w_\ell^T \overline{\phi}_i^{(k)} + b_\ell)}{\sum_{u=1}^L \exp(w_u^T \overline{\phi}_i^{(k)} + b_u)}. \tag{16}$$

The limiting loss, based on the Gaussian mixture (GM) characterization from Theorem 1, is:

$$\mathcal{L}_{\mathrm{GM}}(W, b) := -\sum_{\ell=1}^L \pi_\ell \mathbb{E}_{Y \sim N(\mu_\ell, \Sigma_\ell/\nu_n)} \left[ \log \frac{\exp(w_\ell^T Y + b_\ell)}{\sum_{u=1}^L \exp(w_u^T Y + b_u)} \right]. \tag{17}$$

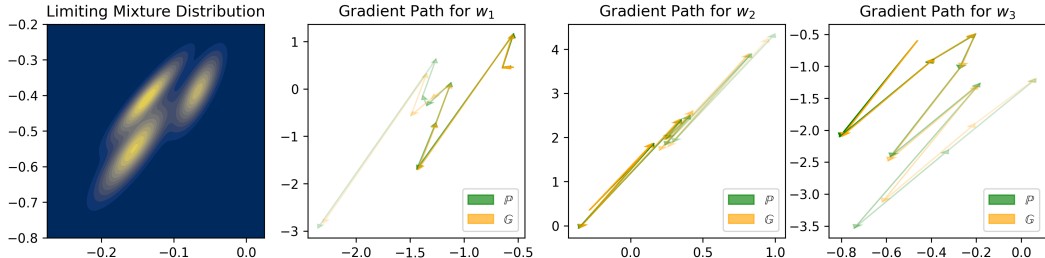

Figure 2: Ten gradient steps of cross-entropy optimization problem for $(A, X)$ drawn from a 3-class CSBM. Shown on the right are gradient paths for samples drawn from empirical and theoretical distributions for $\overline{\phi}^{(k)}$.

For any fixed set of weights $(W, b)$ (e.g., within a ball $\|(W, b)\|_F \leq \mathcal{R}$ for some radius $\mathcal{R}$), the individual loss term for class $\ell$, $\mathrm{CE}_\ell(x; W, b) = -\log \frac{\exp(w_\ell^T x + b_\ell)}{\sum_{u=1}^L \exp(w_u^T x + b_u)}$, is Lipschitz with respect to the feature $x$. This Lipschitz property, combined with the 1-Wasserstein convergence established in Theorem 1 (specifically, the class-conditional form Eq. (11)), leads to the following key result:

**Proposition 1** (Convergence of CE Loss and Gradients). *Under the conditions of Theorem 1, for any fixed radius $\mathcal{R} > 0$:*

*(a) The empirical CE loss converges uniformly to the limiting GM CE loss:*

$$\lim_{n \to \infty} \mathbb{E} \left[ \sup_{\|(W,b)\|_F \leq \mathcal{R}} |\mathfrak{L}_{emp}(W, b) - \mathfrak{L}_{GM}(W, b)| \right] = 0. \tag{18}$$

*(b) The gradients of the empirical CE loss converge uniformly to the gradients of the limiting GM CE loss:*

$$\lim_{n \to \infty} \mathbb{E} \left[ \sup_{\|(W,b)\|_F \leq \mathcal{R}} \left\| \nabla_{(W,b)} \mathfrak{L}_{emp}(W, b) - \nabla_{(W,b)} \mathfrak{L}_{GM}(W, b) \right\|_F \right] = 0. \tag{19}$$

*Consequently, the sequence of parameters $(W_{emp}^*, b_{emp}^*)$ minimizing $\mathfrak{L}_{emp}(W, b)$ within the ball converges in probability to the parameters $(W_{GM}^*, b_{GM}^*)$ minimizing $\mathfrak{L}_{GM}(W, b)$ within the same ball, assuming uniqueness of the minimizer for the limiting problem.*

The proof of (b) relies on the fact that the gradients $\nabla_x \mathrm{CE}_\ell(x; W, b)$ are also Lipschitz in $x$ for bounded $(W, b)$. Proposition 1 formalizes the intuition that training a Poly-GNN with CE loss is asymptotically equivalent to performing CE optimization directly on the identified Gaussian mixture. This explains why gradient descent paths on the empirical loss track those on the limiting GM loss, as illustrated in Figure 2.

The stationarity conditions for optimization problem $\mathfrak{L}_{\mathrm{GM}}(W, b)$ reveal a moment-matching structure:

$$\pi_\ell \mu_\ell = \sum_{u=1}^L \pi_u \mathbb{E}_{Y \sim N(\mu_u, \Sigma_u/\nu_n)}[Y \cdot \widehat{p}_\ell], \quad \text{and} \quad \pi_\ell = \sum_{u=1}^L \pi_u \mathbb{E}_{Y \sim N(\mu_u, \Sigma_u/\nu_n)}[\widehat{p}_\ell], \tag{20}$$

for all $\ell \in [L]$, where $\widehat{p}_\ell := \widehat{p}_\ell(Y; W, b) = \exp(w_\ell^T Y + b_\ell)/\sum_j \exp(w_j^T Y + b_j)$. It is important to note that while the GNN training process converges to this CE solution on the GM, this solution is not necessarily the Bayes optimal classifier for the Gaussian mixture itself (which would be a Quadratic Discriminant Analysis, QDA, classifier). Figure 3 illustrates this, showing that even for large $n$ where $\overline{\phi}^{(k)}$ closely follows the GM, the linear CE boundary can differ from the optimal QDA boundary.

## 4.2 A Precise Mechanism for GNN Oversmoothing

The oversmoothing phenomenon, where GNN performance degrades with depth $k$, is a well-documented empirical observation [5; 17]. Existing explanations often invoke a power iteration argument on a fixed graph matrix, demonstrating that class means collapse towards a common

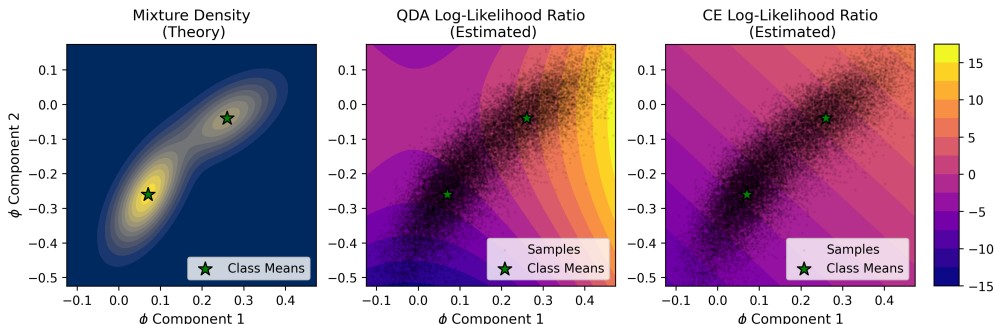

Figure 3: Classifier comparison for data which is 2-dimensional CSBM. On the left is the theoretical density of the 2-class CSBM. The two right plots are the estimated log-likelihood ratios for the QDA and CE estimator respectively. The slight bend in the data is correctly captured by the QDA estimator.

subspace. Our analysis, grounded in a stochastic graph model, provides a fundamentally deeper mechanism.

More precisely, we show that in the sparse, large-graph limit, it is not just the means that collapse. The initial, potentially class-separating covariance of the features vanishes, and is replaced by a purely graph-induced covariance $\Sigma_\ell$ that itself collapses. As our analytical forms show, this new covariance aligns its principal directions with the very same 1-D subspace occupied by the class means. This alignment of signal and noise is a much stronger form of oversmoothing, as it guarantees that the variance concentrates in the same direction as the means, maximally hindering their separability.

To see this, recall the expressions from Eqs. (8) and (9):

$$\mu_\ell = (J^k M)^T e_\ell,$$
$$\Sigma_\ell = (J^{k-1} M)^T \operatorname{diag}(e_\ell^T J)(J^{k-1} M).$$

Consider the symmetric matrix $J_{\text{sym}} = \Pi^{1/2} B \Pi^{1/2}$, which is similar to $J$ (since $J = B\Pi = \Pi^{-1/2} J_{\text{sym}} \Pi^{1/2}$). Let $J_{\text{sym}} = Q \Lambda Q^T$ be its eigendecomposition, with $Q$ orthogonal and $\Lambda = \operatorname{diag}(\lambda_1, \dots, \lambda_L)$ containing the eigenvalues, ordered by magnitude $|\lambda_1| \geq |\lambda_2| \geq \dots$. Then $J^k = \Pi^{-1/2} Q \Lambda^k Q^T \Pi^{1/2}$. If there is a dominant eigenvalue $\lambda_1$ (i.e., $|\lambda_1| > |\lambda_2|$), then for large $k$, the matrix $\Lambda^k \approx \operatorname{diag}(\lambda_1^k, 0, \dots, 0)$. This implies $J^k \approx \lambda_1^k (\Pi^{-1/2} q_1)(q_1^T \Pi^{1/2})$, where $q_1$ is the leading eigenvector of $J_{\text{sym}}$. Let $u_1 = \Pi^{-1/2} q_1$ (a right eigenvector of $J$) and $v_1^T = q_1^T \Pi^{1/2}$ (a left eigenvector of $J$). Then $J^k \approx \lambda_1^k u_1 v_1^T$.

Substituting this into the expressions for $\mu_\ell$ and $\Sigma_\ell$:

- **Class Means:** $\mu_\ell \approx \lambda_1^k (u_1 v_1^T M)^T e_\ell = \lambda_1^k (M^T v_1)(u_1^T e_\ell)$. This shows that for large $k$, all mean vectors $\mu_\ell$ become approximately proportional to the fixed vector $M^T v_1 = M^T \Pi^{1/2} q_1$. The specific proportionality constant $(u_1^T e_\ell)$ depends on the class $\ell$, but the direction is shared.

- **Class Covariances:** Similarly, $J^{k-1} \approx \lambda_1^{k-1} u_1 v_1^T$. Then $\Sigma_\ell \approx \lambda_1^{2(k-1)} (M^T v_1)(\text{scalar}_\ell)(v_1^T M)$, where $\text{scalar}_\ell = u_1^T \operatorname{diag}(e_\ell^T J) u_1$. This indicates that $\Sigma_\ell$ (and thus $\Sigma_\ell / \nu_n$) becomes approximately rank-one, with its dominant direction also aligned with $M^T v_1$.

This power iteration effect driven by $J^k$ and $J^{k-1}$ is the core of the oversmoothing mechanism:

1. **Mean Collapse:** The mean vectors $\mu_\ell$ for different classes tend to align along a common direction $M^T \Pi^{1/2} q_1$. While their magnitudes might differ (scaled by $\lambda_1^k (u_1^T e_\ell)$), their angular separation diminishes. If the initial feature means $M$ projected onto $v_1$ do not maintain sufficient separation, or if $u_1^T e_\ell$ values are too similar across classes, the means become indistinguishable.

2. **Covariance Collapse and Alignment:** The covariance matrices $\Sigma_\ell$ also become rank-deficient and align their principal direction with the same direction as the means.

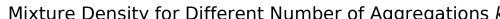

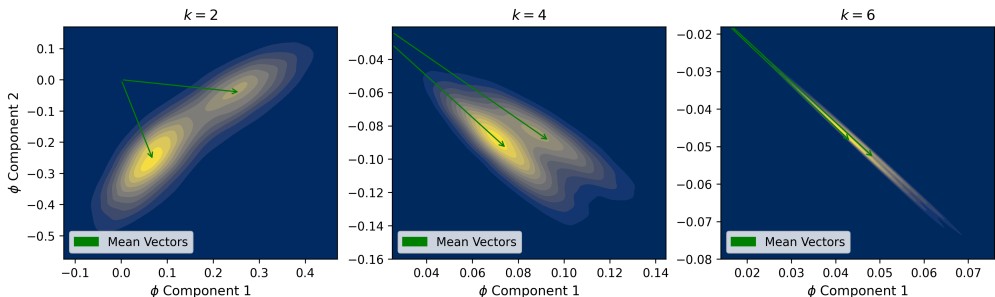

Figure 4: Estimated kernel density plots of the aggregated features $\overline{\phi}^{(k)}$ of a 2-class CSBM at different features depths $k$. A feature collapse in the mean vectors and the class covariances is visible by $k = 4$ and $k = 6$.

The net effect is that the $L$ Gaussian components $N(\mu_\ell, \Sigma_\ell/\nu_n)$ of the feature distribution $\overline{\phi}_i^{(k)}$ effectively collapse onto a 1-dimensional subspace. Within this subspace, they become a mixture of 1-D Gaussians. If the projected means are not well-separated relative to the projected variances along this single dimension, classification becomes extremely difficult, regardless of the original dimensionality $d$ or the initial separability of $M$. This phenomenon is illustrated empirically in Figure 4, where increasing $k$ leads to feature distributions that are elongated along a common axis and overlap significantly. The parameter $\nu_n$ helps shrink the variances overall, but does not prevent this directional collapse induced by $k$.

## 5   Conclusion

We conducted a rigorous asymptotic analysis of $k$-layer Polynomial GNN (Poly-GNN) embeddings on large, sparse, community-based graphs, establishing Central Limit Theorems that precisely characterize their limiting distributions. We showed that degree-normalized features $\overline{\phi}_i^{(k)}$, jointly with labels $z_i$, converge in $W_1$-distance to a Gaussian mixture $N(\mu_\ell, \Sigma_\ell/\nu_n)$ per class $\ell$. We provided exact forms for $\mu_\ell = (J^k M)^T e_\ell$ and $\Sigma_\ell = (J^{k-1}M)^T \operatorname{diag}(e_\ell^T J)(J^{k-1}M)$, determined by initial means $M$, layers $k$, and community interaction matrix $J$. Centered-and-scaled features $\xi_i^{(k)}$ similarly converge to $\sum \pi_\ell N(0, \Sigma_\ell)$.

These findings have key implications. First, training linear classifiers on $\overline{\phi}_i^{(k)}$ with cross-entropy loss is asymptotically equivalent to optimizing on this limiting Gaussian mixture, with uniform convergence of the loss, gradient path, and optimal weights. This theoretically grounds the training behavior of GNN-based classifiers. Second, our explicit characterization of $\mu_\ell$ and $\Sigma_\ell$ offers a clear and nuanced understanding of the GNN oversmoothing phenomenon. The repeated multiplication by the matrix $J$ (to powers $k$ and $k - 1$) acts as a power iteration, causing both the mean vectors and the principal directions of the covariance matrices to align with a low-dimensional (often 1-D) subspace dictated by the leading eigenvectors of $J$. This results in a degenerate, poorly separated Gaussian mixture, thereby diminishing the discriminative power of the GNN embeddings, irrespective of the initial feature dimensionality.

For future work, our framework suggests several avenues. A direct extension would be to extend to degree-corrected stochastic block models (DCSBMs), where we expect a similar CLT to hold provided the normalized degree distribution is stable. Extending the analysis to polynomial filters of the form $\sum_k c_k (A/\nu_n)^k X$ appears feasible, though it would require careful book-keeping of the cross-correlations between different powers of $A$. A more significant challenge, likely requiring new tools beyond our walk-based moment analysis, is the extension to GNNs with non-linear activations or attention mechanisms. As a potential starting point, one could take inspiration from the loss landscape analysis of [8], which applies a walk decomposition to the feed-forward architecture of a fully-connected ReLU network.

## Acknowledgments

This material is based upon work supported by the National Science Foundation under Award No. 1945667.

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

# A Detailed Proofs of Main Theorems

This appendix provides the detailed proofs for Theorem 1 and Theorem 2 presented in Section 3.1. The general proof strategy follows the outline given in Section 3.2.

Before proceeding with the proofs, we establish some notation used throughout the appendices. For a probability measure $\mu$ on $\mathbb{R}^d$ and a vector $\theta \in \mathbb{R}^d$, we denote by $\mu_\theta$ the $\theta$-*projection* (or $\theta$-section) of $\mu$. This is the pushforward measure of $\mu$ under the map $x \mapsto \langle x, \theta \rangle = x^T \theta$. If $X \sim \mu$, then $\mu_\theta$ is the distribution of the real-valued random variable $\langle X, \theta \rangle$. For a measure $\nu$ on $\mathbb{R}$, its $r$-th moment is denoted by $m_r(\nu) = \int x^r d\nu(x)$. For a measure $\mu$ on $\mathbb{R}^d$, its $r$-th absolute moment is $M_r(\mu) = \int \|x\|_2^r d\mu(x)$. The empirical measure of a set of $N$ points $\{Y_i\}_{i=1}^N$ is $\frac{1}{N} \sum_{i=1}^N \delta_{Y_i}$. We denote the expectation of a random empirical measure $\mathbb{P}_n$ as $\overline{\mathbb{P}}_n$, defined by its action on test functions $f$: $\overline{\mathbb{P}}_n[f] = \mathbb{E}[\mathbb{P}_n[f]]$. We use $\mathrm{Lip}(R)$ to denote the class of $R$-Lipschitz functions $f : \mathbb{R}^d \to \mathbb{R}$. Other notation, if not standard, will be defined as it appears.

## A.1 Proof of Theorem 2 (CLT for Centered and Scaled Features)

The proof of Theorem 2 largely follows the structure laid out in Section 3.2. First, we define the key components of the features. Recall the definition of the centered and scaled features from Eq. (3):

$$\xi_i^{(k)} = \sqrt{\nu_n} \Big( \frac{\phi_i^{(k)}}{\nu_n^k} - \mathbb{E}\Big[ \frac{\phi_i^{(k)}}{\nu_n^k} \Big] \Big) = \frac{\phi_i^{(k)} - \mathbb{E}[\phi_i^{(k)}]}{\nu_n^{k-1/2}}.$$

The term $\phi_i^{(k)} - \mathbb{E}[\phi_i^{(k)}]$ represents the deviation of the $i$-th node's $k$-layer feature from its expectation. This deviation can be decomposed as follows. Let $\phi^{(k)}$ be the $n \times d$ matrix of all features.

$$\begin{aligned} \phi^{(k)} - \mathbb{E}[\phi^{(k)}] &= A^k X - \mathbb{E}[A^k X] \\ &= A^k X - \mathbb{E}[A^k]\mathbb{E}[X] \quad \text{(since $A$ and $X$ are independent given $z$)} \\ &= (A^k - \mathbb{E}[A^k])X + \mathbb{E}[A^k](X - \mathbb{E}[X]) \\ &=: \mathring{\Delta} + \mathring{\Lambda}. \end{aligned} \tag{21}$$

Here, $\mathring{\Delta} = (A^k - \mathbb{E}[A^k])X$ captures the randomness from the graph structure $A$, and $\mathring{\Lambda} = \mathbb{E}[A^k](X - \mathbb{E}[X])$ captures the randomness from the initial node features $X$ (around their means). Both $\mathring{\Delta}$ and $\mathring{\Lambda}$ are $n \times d$ matrices. Let $\mathring{\Delta}_i$ and $\mathring{\Lambda}_i$ denote their $i$-th rows (viewed as $d \times 1$ column vectors for consistency with $\xi_i^{(k)}$).

We define the normalized versions:

$$\Delta_i := \mathring{\Delta}_i / \nu_n^{k-1/2}, \quad \Lambda_i := \mathring{\Lambda}_i / \nu_n^{k-1/2}.$$

With this notation, the centered and scaled feature for node $i$ is:

$$\xi_i^{(k)} = \Delta_i + \Lambda_i.$$

For any projection vector $\theta \in S^{d-1}$, we denote $\Delta_{i,\theta} = \langle \Delta_i, \theta \rangle = \Delta_i^T \theta$ and $\Lambda_{i,\theta} = \langle \Lambda_i, \theta \rangle = \Lambda_i^T \theta$.

Now, we aim to show $\mathbb{E}[W_1(\mathbb{P}_n, \mathbb{G})] \to 0$ where $\mathbb{P}_n = \frac{1}{n} \sum_{i=1}^n \delta_{\xi_i^{(k)}}$. This is achieved by showing:

1. $\mathbb{E}[W_1(\mathbb{P}_n, \overline{\mathbb{P}}_n)] \to 0$, where $\overline{\mathbb{P}}_n = \mathbb{E}[\mathbb{P}_n]$.
2. $W_1(\overline{\mathbb{P}}_n, \mathbb{G}) \to 0$.

**Part 1: Concentration of $\mathbb{P}_n$ around $\overline{\mathbb{P}}_n$.** This part relies on Proposition 8 (from Appendix D). To apply Proposition 8, we need to verify its conditions for $Y_{i,n} = \xi_i^{(k)}$:

(a) *Uniform $\Psi_{r_n}$ sub-Gaussianity of projections*: For any $\theta \in S^{d-1}$, $\{\langle \xi_i^{(k)}, \theta \rangle\}_{i=1}^n$ are uniformly $\Psi_{r_n}$ sub-Gaussian (see Appendix C for the definition):

**Lemma 1.** $\sup_{i \in [n]} \|\langle \xi_i^{(k)}, \theta \rangle\|_{\Psi_{r_n}} \lesssim C(\sigma, x_*)$ *for all* $\theta \in S^{d-1}$.

*Proof.* Combining Propositions 2 and 3 (from Appendix A.3), we have

$$\|\langle \xi_i^{(k)}, \theta \rangle\|_{\Psi_{r_n}} \lesssim \|\Delta_{i,\theta}\|_{\Psi_{r_n}} + \|\Lambda_{i,\theta}\|_{\Psi_{r_n}} \lesssim \kappa_0 + \sigma \beta_{k,n}$$

where $\kappa_0$ is a constant only dependent on $\sigma$ and $x_*$, and $\beta_{k,n} = o(1)$. The result follows. $\square$

(b) *Variance of empirical moments*: We have the following result:

**Lemma 2.** $\lim_{n \to \infty} Var\left( n^{-1} \sum_i \langle \xi_i^{(k)}, \theta \rangle^r \right) = 0$ *for all* $r \in \mathbb{N}$ *and* $\theta \in S^{d-1}$.

*Proof.* By Lemma 5, $\lim_{n \to \infty} \|\frac{1}{n} \sum_i \langle \xi_i^{(k)}, \theta \rangle^r - \frac{1}{n} \sum_{i=1}^n \Delta_{i,\theta}^r\|_{L^2} = 0$. Lemma 6 gives $Var(n^{-1} \sum_i \Delta_{i,\theta}^r) \lesssim n^{-1}$. The result follows from the inequality $var(A) \leq 6\|A - B\|_{L^2}^2 + 3 \, var(B)$ for any random variables $A$ and $B$ in $L^2$. $\square$

(c) *Uniformly bounded first moment of* $\overline{\mathbb{P}}_n$: $\sup_{n \geq 1} M_1(\overline{\mathbb{P}}_n) < \infty$. This follows since by Proposition 4 $m_1(\overline{\mathbb{P}}_{n,\theta})$ converge to $m_1(\mathbb{G}_\theta)$, which is finite, for all $\theta \in S^{d-1}$. The claims then follows from Proposition 7.

With these conditions met, Proposition 8 (from Appendix D) implies $\mathbb{E}[W_1(\mathbb{P}_n, \overline{\mathbb{P}}_n)] \to 0$.

**Part 2: Convergence of** $\overline{\mathbb{P}}_n$ **to** $\mathbb{G}$. Proposition 4 (from Appendix A.4) shows $m_r(\overline{\mathbb{P}}_{n,\theta}) \to m_r(\mathbb{G}_\theta)$ for all $\theta$ and $r$. Since $\mathbb{G}_\theta$ is a mixture of Gaussians, it is determined by its moments. This implies weak convergence $\overline{\mathbb{P}}_{n,\theta} \rightsquigarrow \mathbb{G}_\theta$. The convergence of moments also implies uniform integrability of all moments for $\{\overline{\mathbb{P}}_{n,\theta}\}_{n \geq 1}$. This, combined with weak convergence, yields $W_1(\overline{\mathbb{P}}_{n,\theta}, \mathbb{G}_\theta) \to 0$ for all $\theta \in S^{d-1}$ (e.g., by [1, Proposition 7.1.5]). Proposition 8 (from Appendix) outlines the conditions necessary to lift this claim to $W_1(\overline{\mathbb{P}}_n, \mathbb{G}) \to 0$. Here, we note that condition (c), $\sup_{n \geq 1}(M_1(\overline{\mathbb{P}}_n) < \infty$, is satisfied because $M_1(\overline{\mathbb{P}}_n)$ is uniformly bounded (as argued in Part 1c).

The class-conditional convergence statement in Theorem 2 follows from a similar argument by considering per-class empirical measures $\mathbb{P}_{n,\ell}$ and their expectations $\overline{\mathbb{P}}_{n,\ell}$, and showing their convergence to $N(0, \Sigma_\ell)$. See the proof of Proposition 5 (a restatement of the class-conditional convergence) for details.

## A.2 Proof of Theorem 1 (CLT for Degree-Normalized Features and Labels)

The proof of Theorem 1 closely mirrors that of Theorem 2, with adjustments for the non-zero means and the $\nu_n^{-1}$ scaling in the covariance. Let $\widetilde{\mathbb{P}}_n^{joint}$ be the empirical measure of $(z_i, \overline{\phi}_i^{(k)})$ and $\mathbb{G}_n^{joint}$ be its target limit. The convergence in $W_1$ can be established by showing convergence of expectations of Lipschitz functions $f(z, x)$. The core argument involves showing that for $i \in \mathcal{C}_\ell$, $\overline{\phi}_i^{(k)}$ behaves like a draw from $N(\mu_\ell, \Sigma_\ell/\nu_n)$.

1. **Mean Convergence:** Lemma 3 establishes that $\mathbb{E}[\overline{\phi}_i^{(k)}]$ converges to a general limit $\gamma_i$.

   **Lemma 3.** *Define limiting mean* $\gamma_i = e_i^T(\mathbb{E}[A/\nu_n])^k \mathbb{E}[X]$. *Assume Assumption 4 and suppose* $\nu_n \geq 1$. *Then,*

   $$\max_{i \in [n]} \|\mathbb{E}[\overline{\phi}_i^{(k)}] - \gamma_i^T\|_2 \leq C(k) \, x_* \, \nu_n^{-1}.$$

   Under the specific community-based graph model, this general limit $\gamma_i$ further simplifies for nodes within a class $\mathcal{C}_\ell$ to the class-specific mean $\mu_\ell$, as stated in the following lemma.

   **Lemma 4.** *Under the conditions of Theorem 1 (which include the CSBM structure and Assumptions 1–4), let* $\gamma_i^T = e_i^T(\mathbb{E}[A/\nu_n])^k \mathbb{E}[X]$ *be the* $1 \times d$ *row vector defined in Lemma 3. For any node* $i \in \mathcal{C}_\ell$, *its limiting mean* $\gamma_i^T$ *converges to* $\mu_\ell^T = e_\ell^T J^k M$. *More precisely,*

   $$\max_{\ell \in [L]} \sup_{i \in \mathcal{C}_\ell} \|\gamma_i^T - \mu_\ell^T\|_2 = o(1)$$

   *Proof.* The expected adjacency matrix of an undirected, loop-less SBM is $\mathbb{E}[A] = (\nu_n/n)(P - \text{diag}(P))$, where $P = ZBZ^T$ and $\text{diag}(P)$ contains the diagonal entries of $P$. Thus, $\mathbb{E}[A/\nu_n] =$

$(P/n) - (\text{diag}(P)/n)$. The difference between $(\mathbb{E}[A/\nu_n])^k$ and $(P/n)^k$ can be bounded. Since matrix exponentiation $H \mapsto H^k$ is locally Lipschitz for matrices with bounded operator norm (which $(P/n)$ and $\mathbb{E}[A/\nu_n]$ are, as their entries are $O(1/n)$ and norms are $O(1)$), and

$$\|\mathbb{E}[A/\nu_n] - P/n\|_{\text{op}} = \|\text{diag}(P)/n\|_{\text{op}} = \max_{\ell' \in [L]} n^{-1} B_{\ell' \ell'} = O(n^{-1}), \tag{22}$$

it follows that $\|(\mathbb{E}[A/\nu_n])^k - (P/n)^k\|_{\text{op}} = O(n^{-1})$. Given $\mathbb{E}[X] = ZM$ has bounded row norms (from Assumption 4), we can write:

$$\gamma_i^T = e_i^T (P/n)^k \mathbb{E}[X] + e_i^T ((\mathbb{E}[A/\nu_n])^k - (P/n)^k) \mathbb{E}[X]$$
$$= e_i^T (P/n)^k ZM + O(n^{-1}) \cdot \|e_i^T\|_{\text{op}} \|\mathbb{E}[X]\|_{\text{op}}.$$

Since $\|e_i^T\|_{\text{op}} = 1$ and $\|\mathbb{E}[X]\|_{\text{op}}$ is bounded (e.g., by $\sqrt{L} \max_\ell \|M_{\ell,\cdot}\|_2 \leq \sqrt{L} x_*$), the error term is $O(n^{-1})$. So,

$$\gamma_i^T = e_i^T (P/n)^k ZM + O(n^{-1}).$$

Now consider the main term $e_i^T (P/n)^k ZM$. For a node $i \in \mathcal{C}_\ell$, we have $e_i^T Z = e_\ell^T$ (where $e_i \in \mathbb{R}^n, e_\ell \in \mathbb{R}^L$).

$$e_i^T (P/n)^k ZM = e_i^T (ZBZ^T/n)^k ZM$$
$$= e_i^T Z(B(Z^T Z/n))^k M$$
$$= e_\ell^T (B\widetilde{\Pi})^k M \quad (\text{since } Z^T Z/n = \text{diag}(|\mathcal{C}_s|/n)_{s=1}^L = \widetilde{\Pi})$$
$$= e_\ell^T \widetilde{J}^k M,$$

where $\widetilde{J} = B\widetilde{\Pi}$ and $\widetilde{\Pi} = \text{diag}(\widetilde{\pi}_1, \ldots, \widetilde{\pi}_L)$ with $\widetilde{\pi}_s = |\mathcal{C}_s|/n$. We are given $\mu_\ell^T = e_\ell^T J^k M$, where $J = B\Pi$. The difference is $e_\ell^T (\widetilde{J}^k - J^k) M$.

From the Assumption 3, $\widetilde{\pi}_s = \pi_s + o(1)$, which implies $\widetilde{\Pi} = \Pi + E_n$ where $E_n$ is a diagonal matrix with entries $o(1)$. Thus, $\|\widetilde{\Pi} - \Pi\|_{\text{op}} = o(1)$. Then, $\widetilde{J} - J = B(\widetilde{\Pi} - \Pi) = BE_n$. So, $\|\widetilde{J} - J\|_{\text{op}} \leq \|B\|_{\text{op}} \|E_n\|_{\text{op}} = O(1) \cdot o(1) = o(1)$. Using the identity $A^k - B^k = \sum_{j=0}^{k-1} A^j (A - B) B^{k-1-j}$, and since $\|J\|_{\text{op}}$ and $\|\widetilde{J}\|_{\text{op}}$ are $O(1)$ (as $\|B\|_{\text{op}}$ and $\|\Pi\|_{\text{op}}$ are $O(1)$),

$$\|\widetilde{J}^k - J^k\|_{\text{op}} \leq k \cdot \max(\|J\|_{\text{op}}, \|\widetilde{J}\|_{\text{op}})^{k-1} \cdot \|\widetilde{J} - J\|_{\text{op}} = O(1) \cdot o(1) = o(1).$$

Therefore,

$$\|e_\ell^T (\widetilde{J}^k - J^k) M\|_2 \leq \|e_\ell^T\|_{\text{op}} \|\widetilde{J}^k - J^k\|_{\text{op}} \|M\|_{\text{op}} = 1 \cdot o(1) \cdot O(1) = o(1).$$

Combining the two error terms:

$$\gamma_i^T = e_\ell^T J^k M + o(1) + O(n^{-1}).$$

Since $\nu_n = o(n)$ (Assumption 2), $n^{-1} = o(\nu_n^{-1})$ which is also $o(1)$. Thus, the dominant error term is $o(1)$. The bounds are uniform over $i \in \mathcal{C}_\ell$ and $\ell \in [L]$ because the operator norm bounds on $B, \Pi, M$ and the rate of convergence in Assumption 3 are uniform. $\qquad \square$

2. **Covariance Characterization:** The deviation $\overline{\phi}_i^{(k)} - \mathbb{E}[\overline{\phi}_i^{(k)}] = \xi_i^{(k)}/\sqrt{\nu_n}$. The analysis for $\xi_i^{(k)}$ (specifically, the characterization of its moments leading to Proposition 4 in Appendix A.4) shows its asymptotic covariance, conditional on $z_i = \ell$, is $\Sigma_\ell$. Thus, the covariance of $\overline{\phi}_i^{(k)} - \mathbb{E}[\overline{\phi}_i^{(k)}]$ (and asymptotically, of $\overline{\phi}_i^{(k)} - \mu_\ell$ for $i \in \mathcal{C}_\ell$) is $\Sigma_\ell/\nu_n$.

3. **Moment Matching and Concentration:** Similar to Theorem 2, one shows that the moments of $(\overline{\phi}_i^{(k)} - \mu_\ell)$ (for $i \in \mathcal{C}_\ell$), when appropriately scaled, match those of $N(0, \Sigma_\ell/\nu_n)$. Concentration arguments analogous to Part 1 of Theorem 2's proof apply.

Steps 2 and 3 above are rigorously formalized during the proof of the class-conditional version of the statement (Eq. (11) which is stated and proved as Proposition 6 in Appendix A.4).

### A.3  Supporting Lemmas for Moment Analysis

We borrow the following two key results from [18]:

**Proposition 2.** *Suppose $X_i - \mathbb{E}[X_i] \sim \mathrm{SG}(\sigma^2)$ and $\nu_n \geq 1$. Then, for $\Lambda_{i,\theta} = \langle \mathring{\Lambda}_i / \nu_n^{k-1/2}, \theta \rangle$:*

$$\|\Lambda_{i,\theta}\|_{\Psi_{r_n}} \lesssim \sigma \beta_{k,n} \quad where \quad \beta_{k,n} = (\nu_n^{-k+1})^{1/2} \cdot \mathbb{1}\{k \text{ is even}\} + (\nu_n/n)^{1/2}.$$

*Proof.* A component-wise version of the above (i.e. with $\theta$ a coordinate basis vector) is proven in [18, Section 4.1]. The general case follows by a similar argument. Broadly, this follows from the fact that $\mathring{\Lambda}_i = \mathbb{E}[A^k](X - \mathbb{E}[X])/\nu_n^{k-1/2}$ is a linear transformation of sub-Gaussian random vectors $\{X_{i*} - \mathbb{E}[X_{i*}]\}_i$. The rate $\beta_{k,n}$ is obtained through path counting on $\mathbb{E}[A^k]_{ij}$ for each $j \in [n]$. $\square$

**Proposition 3.** *Let $t_* = rk - \lceil r/2 \rceil$ and $\kappa_0 = 4\max\{C_1\sigma, x_*\}$ for a distribution-dependent constant $C_1$. Assume Assumption (4) and suppose $\nu_n \geq 1$. For $\epsilon \in [0, 1]$, let*

$$r_n(\epsilon) := \max\{r \in 2\mathbb{N} : 3(\kappa_0 rke^k)^r \leq \nu_n^{1-\epsilon}\}. \tag{23}$$

*Then, for $\mathring{\Delta}_{i,\theta} = \langle (A^k - \mathbb{E}[A^k])X, \theta \rangle$ and for all $r \leq r_n(0)$:*

$$\mathbb{E}|\mathring{\Delta}_{i,\theta}|^r \leq 2(\sqrt{r}\kappa_0)^r \nu_n^{t_*}. \tag{24}$$

*As a consequence, for $\Delta_{i,\theta} = \mathring{\Delta}_{i,\theta}/\nu_n^{k-1/2}$, we have $\|\Delta_{i,\theta}\|_{\Psi_{r_n}} \lesssim \kappa_0$.*

*Proof.* This result is proven in [18]. The power $\nu_n^{t_*}$ arises from counting dominant walk structures contributing to the $r$-th moment. $\square$

With these propositions place, we show that, in the sparse setting, the sliced moments of $\langle \xi_i^{(k)}, \theta \rangle$ are determined by the moments of graph noise $\Delta_{i,\theta}$. That is to say, as $n$ grows large and the graph grows sparse, the contribution of feature noise $\Lambda_{i,\theta}$ to our normalized features $\langle \xi_i^{(k)}, \theta \rangle$ is negligible. This has important downstream consequences to our limiting aggregated features, as it implies the feature noise covariance will not appear in the final limiting form of the aggregated feature covariance.

**Lemma 5.** *Assume Assumptions 1, 2, and 4. For any $r \in \mathbb{N}$:*

$$\lim_{n \to \infty} \max_{i \in [n]} \left\| \langle \xi_i^{(k)}, \theta \rangle^r - \Delta_{i,\theta}^r \right\|_{L^2} = 0.$$

*Proof.* Using the decomposition $\langle \xi_i^{(k)}, \theta \rangle = \Delta_{i,\theta} + \Lambda_{i,\theta}$, we have

$$\langle \xi_i^{(k)}, \theta \rangle^r - \Delta_{i,\theta}^r = \sum_{s=1}^{r} \binom{r}{s} \Delta_{i,\theta}^{r-s} \Lambda_{i,\theta}^s.$$

By Minkowski inequality (for $L_2$ norm of sums):

$$\left\| \sum_{s=1}^{r} \binom{r}{s} \Delta_{i,\theta}^{r-s} \Lambda_{i,\theta}^s \right\|_{L^2} \leq \sum_{s=1}^{r} \binom{r}{s} \left\| \Delta_{i,\theta}^{r-s} \Lambda_{i,\theta}^s \right\|_{L^2}$$

By Hölder inequality, with $1/p = (r-s)/r$ and $1/q = s/r$,

$$\left\| \Delta_{i,\theta}^{r-s} \Lambda_{i,\theta}^s \right\|_{L^2}^2 = \mathbb{E}[\Delta_{i,\theta}^{2(r-s)} \Lambda_{i,\theta}^{2s}] \leq (\mathbb{E}\Delta_{i,\theta}^{2r})^{1-s/r} (\mathbb{E}\Lambda_{i,\theta}^{2r})^{s/r}.$$

This is $\|\Delta_{i,\theta}\|_{L^{2r}}^{2(r-s)} \cdot \|\Lambda_{i,\theta}\|_{L^{2r}}^{2s}$. For $n$ large enough so $2r \leq r_n$, Proposition 3 (via Lemma 9) gives $\|\Delta_{i,\theta}\|_{L^{2r}} \lesssim \kappa_0\sqrt{2r}$. Proposition 2 (via Lemma 9) gives $\|\Lambda_{i,\theta}\|_{L^{2r}} \lesssim \sigma\beta_{k,n}\sqrt{2r}$. Take $n$ large enough so that $\beta_{k,n} \leq 1$. Then, for $s \geq 1$, we have $\beta_{k,n}^{2s} \leq \beta_{k,n}$, hence $\|\Lambda_{i,\theta}\|_{L^{2r}}^{2s} \lesssim \sigma^{2s}\beta_{k,n}(2r)^s$. Since $\beta_{k,n} \to 0$ from Proposition 2 (as $\nu_n \to \infty$, $\nu_n = o(n)$), and all other terms are bounded, the sum tends to 0. The convergence is uniform over $i$ as the bounds are uniform. $\square$

**Lemma 6.** *Under Assumption 1 and 4. For every* $i, i' \in [n]$, $r \in \mathbb{N}$ *and* $\theta \in S^{d-1}$,

$$\mathrm{Cov}\big(\Delta_{i,\theta}^r, \Delta_{i',\theta}^r\big) \lesssim n^{-1\{i \neq i'\}}.$$

*In particular,* $\mathrm{Var}\big(n^{-1} \sum_{i=1}^n \Delta_{i,\theta}^r\big) \lesssim n^{-1}$ *for* $r \in \mathbb{N}$ *and* $\theta \in S^{d-1}$.

The proof of Lemma 6 is quite involved, using combinatorics of walk sequences, and appears in Appendix E.

**Lemma 7.** *Let* $r \in 2\mathbb{N}$ *and* $0 < \epsilon < 1$. *Assume Assumption* (4) *and suppose* $\nu_n \geq 1$. *If* $r \leq r_n(\epsilon)$, *then*

$$\max_{i \in [n]} \big| \mathbb{E}[\Delta_{i,\theta}^r] - (r-1)!! \cdot \widetilde{\sigma}_{i,\theta}^r \big| \leq C(r)\, x_*^r\, \nu_n^{-\epsilon}$$

*where* $\widetilde{\sigma}_{i,\theta}^2 := \|V_i \mathbb{E}[A/\nu_n]^{k-1} \mathbb{E}[X]\theta\|_2^2$ *and* $V_i = \big[\mathrm{diag}\big((p_{i1}(1 - p_{i1}), \ldots, p_{in}(1 - p_{in}))/\nu_n\big]^{1/2}$.

The proof of Lemma 7 appears in Appendix E and involves walk-based proxy term $\widetilde{T}_i^{\mathrm{hi}}(r)$ and careful counting of dominant vs non-dominant walk structures.

**Lemma 8** (Odd Moment Control for $\Delta_{i,\theta}$). *Under Assumptions 1–4, for any odd integer* $r \geq 1$ *and any unit vector* $\theta \in \mathbb{R}^d$,

$$\lim_{n \to \infty} \max_{i \in [n]} |\mathbb{E}[\Delta_{i,\theta}^r]| = 0.$$

*More specifically,* $\mathbb{E}[\Delta_{i,\theta}^r] = O(\nu_n^{-1/2})$.

*Proof.* This follows from Proposition 3. For an odd $r$, the moment bound for $\Delta_{i,\theta}$ is $\mathbb{E}|\Delta_{i,\theta}|^r \lesssim (\sqrt{r}\kappa_0)^r \nu_n^{r/2 - \lceil r/2 \rceil} = (\sqrt{r}\kappa_0)^r \nu_n^{-1/2}$. $\qquad\square$

## A.4 Supporting Results for Specialization to Community-Based Graphs

**Proposition 4.** *Under Assumptions 1–4,*

$$\lim_{n \to \infty} \frac{1}{n} \sum_{i=1}^n \mathbb{E}\langle \xi_i^{(k)}, \theta \rangle^r = (r-1)!! \sum_{\ell=1}^L \pi_\ell \big((J^{k-1}M\theta)^T \mathrm{diag}(e_\ell^T J)(J^{k-1}M\theta)\big)^{r/2} \cdot \mathbb{1}\{r \text{ is even}\}.$$

*Stated differently,* $m_r(\overline{\mathbb{P}}_{n,\theta}) \to m_r(\mathbb{G}_\theta)$ *where* $\mathbb{G}_\theta = \sum_{\ell=1}^L \pi_\ell N(0, \theta^T \Sigma_\ell \theta)$.

*Proof.* We proceed in steps:

*Step 1: Approximate with moments of* $\Delta_{i,\theta}$. By Lemma 5 (specifically, $\|\langle \xi_i^{(k)}, \theta \rangle^r - \Delta_{i,\theta}^r\|_{L^1} \to 0$ since $L_2$ convergence implies $L_1$), we have $\mathbb{E}\langle \xi_i^{(k)}, \theta \rangle^r = \mathbb{E}[\Delta_{i,\theta}^r] + o(1)$, where the $o(1)$ term is uniform over $i$. Thus,

$$\frac{1}{n} \sum_{i=1}^n \mathbb{E}\langle \xi_i^{(k)}, \theta \rangle^r = \frac{1}{n} \sum_{i=1}^n \mathbb{E}[\Delta_{i,\theta}^r] + o(1).$$

*Step 2: Handle odd moments.* If $r$ is an odd integer, by Lemma 8, $\mathbb{E}[\Delta_{i,\theta}^r] = o(1)$ uniformly in $i$. Therefore,

$$\lim_{n \to \infty} \frac{1}{n} \sum_{i=1}^n \mathbb{E}[\Delta_{i,\theta}^r] = 0.$$

This matches the proposition statement, as $\mathbb{1}\{r \text{ is even}\} = 0$ for odd $r$.

*Step 3: Handle even moments using* $\widetilde{\sigma}_{i,\theta}^r$. If $r$ is an even integer, by Lemma 7,

$$\mathbb{E}[\Delta_{i,\theta}^r] = (r-1)!! \cdot \widetilde{\sigma}_{i,\theta}^r + o(1),$$

uniformly in $i$. Here, $\widetilde{\sigma}_{i,\theta}^2 = \|V_i \mathbb{E}[A/\nu_n]^{k-1} \mathbb{E}[X]\theta\|_2^2$. So we need to analyze the limit of $\frac{1}{n} \sum_{i=1}^n (r-1)!! \cdot \widetilde{\sigma}_{i,\theta}^r$.

*Step 4: Analyze $\widetilde{\sigma}^2_{i,\theta}$ under the CSBM structure.* We have

$$\widetilde{\sigma}^2_{i,\theta} = (\mathbb{E}[A/\nu_n]^{k-1}\mathbb{E}[X]\theta)^T V_i^2 (\mathbb{E}[A/\nu_n]^{k-1}\mathbb{E}[X]\theta).$$

where $V_i^2 = \nu_n^{-1}\mathrm{diag}((p_{ij}(1-p_{ij}))_{j=1}^n) = \nu_n^{-1}\mathrm{diag}(e_i^T\mathbb{E}[A])(I_n - \mathrm{diag}(e_i^T\mathbb{E}[A]))$. Since by assumption $\nu_n = o(n)$, we have $e_i^T\mathbb{E}[A] = O(\nu_n/n) = o(1)$ uniformly in $i$. It follows that

$$V_i^2 = \nu_n^{-1}\mathrm{diag}(e_i^T\mathbb{E}[A]) + o(1).$$

Moreover, as shown in the proof of Lemma 4 (specifically Eq. (22)), $\mathbb{E}[A/\nu_n] = P/n + O(n^{-1})$, where $P = ZBZ^T$. Substituting we get

$$\widetilde{\sigma}^2_{i,\theta} = ((P/n)^{k-1}\mathbb{E}[X]\theta)^T\mathrm{diag}(e_i^T P/n)((P/n)^{k-1}\mathbb{E}[X]\theta) + o(1).$$

Under the CSBM structure, we have $\mathbb{E}[X]\theta = ZM\theta$. Similar to the derivation in Lemma 4's proof:

$$(P/n)^{k-1}ZM\theta = Z(B\widetilde{\Pi})^{k-1}M\theta = Z\widetilde{J}_n^{k-1}M\theta.$$

If node $i \in \mathcal{C}_\ell$, then $e_i^T P/n = e_\ell^T(BZ^T/n)$. The term $Z^T\mathrm{diag}(e_i^T P/n)Z$ becomes a diagonal $L \times L$ matrix. For $i \in \mathcal{C}_\ell$:

$$(Z^T\mathrm{diag}(e_i^T P/n)Z)_{s,s'} = \sum_{j=1}^n Z_{js}(e_i^T P/n)_j Z_{js'}$$

$$= \sum_{j\in\mathcal{C}_s, s=s'}(P_{ij}/n) = \sum_{j\in\mathcal{C}_s, s=s'}(B_{z_i z_j}/n)$$

$$= \mathbb{1}\{s = s'\} \cdot (B_{\ell s}|\mathcal{C}_s|/n) = \mathbb{1}\{s = s'\} \cdot B_{\ell s}\widetilde{\pi}_s.$$

So, $Z^T\mathrm{diag}(e_i^T P/n)Z = \mathrm{diag}((B_{\ell s}\widetilde{\pi}_s)_{s=1}^L) = \mathrm{diag}(e_\ell^T\widetilde{J}_n)$. Therefore, for $i \in \mathcal{C}_\ell$:

$$\widetilde{\sigma}^2_{i,\theta} = (\widetilde{J}_n^{k-1}M\theta)^T\mathrm{diag}(e_\ell^T\widetilde{J}_n)(\widetilde{J}_n^{k-1}M\theta) + o(1).$$

Let $\sigma^2_{\ell,\theta}(\widetilde{J}_n) = (\widetilde{J}_n^{k-1}M\theta)^T\mathrm{diag}(e_\ell^T\widetilde{J}_n)(\widetilde{J}_n^{k-1}M\theta)$. This term is the same for all $i \in \mathcal{C}_\ell$ up to $o(1)$ errors.

*Step 5: Averaging over $i$ and taking limits.* For even $r$:

$$\frac{1}{n}\sum_{i=1}^n\mathbb{E}[\Delta^r_{i,\theta}] = \frac{(r-1)!!}{n}\sum_{i=1}^n\widetilde{\sigma}^r_{i,\theta} + o(1)$$

$$= (r-1)!!\sum_{\ell=1}^L\frac{|\mathcal{C}_\ell|}{n}\left(\frac{1}{|\mathcal{C}_\ell|}\sum_{i\in\mathcal{C}_\ell}\widetilde{\sigma}^r_{i,\theta}\right) + o(1)$$

$$= (r-1)!!\sum_{\ell=1}^L\widetilde{\pi}_\ell \cdot (\sigma^2_{\ell,\theta}(\widetilde{J}_n))^{r/2} + o(1).$$

As $n \to \infty$, by Assumption 3, $\widetilde{\pi}_\ell \to \pi_\ell$. Also, $\|\widetilde{J}_n - J\|_{\mathrm{op}} \to 0$ (due to $\widetilde{\Pi} \to \Pi$). Since $\sigma^2_{\ell,\theta}(\cdot)$ is a continuous function of its matrix argument (in terms of matrix entries or operator norm for fixed $M, \theta, B, e_\ell, k$), we have $\sigma^2_{\ell,\theta}(\widetilde{J}_n) \to \sigma^2_{\ell,\theta}(J)$. Let $\sigma^{*2}_{\ell,\theta} = (J^{k-1}M\theta)^T\mathrm{diag}(e_\ell^T J)(J^{k-1}M\theta)$. The limit becomes:

$$(r-1)!!\sum_{\ell=1}^L\pi_\ell(\sigma^{*2}_{\ell,\theta})^{r/2}.$$

This is precisely the $r$-th moment of $\mathbb{G}_\theta = \sum_{\ell=1}^L\pi_\ell N(0,\sigma^{*2}_{\ell,\theta})$. Note that $\sigma^{*2}_{\ell,\theta} = \theta^T\Sigma_\ell\theta$ where $\Sigma_\ell = (J^{k-1}M)^T\mathrm{diag}(e_\ell^T J)(J^{k-1}M)$. The proof is complete. $\qquad\square$

**Proposition 5 (Part of Theorem 2).** *Consider the setting of Proposition 4. Let $\mathbb{G}_\ell = N(0, \Sigma_\ell)$ for $\ell \in [L]$. Then for any $R > 0$:*

$$\mathbb{E}\left\{\sup_{f_1,\ldots,f_L\in\mathrm{Lip}(R)}\left|\frac{1}{n}\sum_{\ell=1}^L\sum_{i\in\mathcal{C}_\ell}f_\ell(\xi_i^{(k)}) - \sum_{\ell=1}^L\pi_\ell\mathbb{E}_{Y\sim\mathbb{G}_\ell}[f_\ell(Y)]\right|\right\} \to 0.$$

*Proof.* Let $\mathbb{P}_{n,\ell} = \frac{1}{|\mathcal{C}_\ell|} \sum_{i \in \mathcal{C}_\ell} \delta_{\xi_i^{(k)}}$ be the class-conditional empirical measure for class $\ell$. Let $f_\ell \in \text{Lip}(R)$. We can assume $f_\ell(0) = 0$ without loss of generality by considering $f_\ell(x) - f_\ell(0)$, as this does not change the difference of expectations for centered measures and preserves the Lipschitz constant. The term we want to show goes to zero is:

$$\Delta_n := \sup_{f_1,\dots,f_L \in \text{Lip}(R)} \left| \sum_{\ell=1}^{L} \frac{|\mathcal{C}_\ell|}{n} \mathbb{P}_{n,\ell}[f_\ell] - \sum_{\ell=1}^{L} \pi_\ell \mathbb{G}_\ell[f_\ell] \right|.$$

Using the triangle inequality:

$$\Delta_n \leq \sup_{f_1,\dots,f_L \in \text{Lip}(R)} \sum_{\ell=1}^{L} \left| \frac{|\mathcal{C}_\ell|}{n} \mathbb{P}_{n,\ell}[f_\ell] - \pi_\ell \mathbb{P}_{n,\ell}[f_\ell] \right|$$

$$+ \sup_{f_1,\dots,f_L \in \text{Lip}(R)} \sum_{\ell=1}^{L} |\pi_\ell \mathbb{P}_{n,\ell}[f_\ell] - \pi_\ell \mathbb{G}_\ell[f_\ell]|$$

$$\leq \sum_{\ell=1}^{L} \left| \frac{|\mathcal{C}_\ell|}{n} - \pi_\ell \right| \sup_{f_\ell \in \text{Lip}(R)} |\mathbb{P}_{n,\ell}[f_\ell]|$$

$$+ \sum_{\ell=1}^{L} \pi_\ell \sup_{f_\ell \in \text{Lip}(R)} |\mathbb{P}_{n,\ell}[f_\ell] - \mathbb{G}_\ell[f_\ell]|.$$

The second term is $\sum_{\ell=1}^{L} \pi_\ell R \cdot W_1(\mathbb{P}_{n,\ell}, \mathbb{G}_\ell)$ by definition of $W_1$ (scaled by $R$). Let $T_{1,n}$ and $T_{2,n}$ be the two terms.

For $T_{1,n}$: Since $f_\ell(0) = 0$ and $f_\ell \in \text{Lip}(R)$, $|\mathbb{P}_{n,\ell}[f_\ell]| \leq \mathbb{P}_{n,\ell}[|f_\ell(x)|] \leq R \cdot \mathbb{P}_{n,\ell}[\|x\|]$. So, $\mathbb{E}[\sup_{f_\ell \in \text{Lip}(R)} |\mathbb{P}_{n,\ell}[f_\ell]|] \leq R \cdot \mathbb{E}[\mathbb{P}_{n,\ell}[\|x\|]] = R \cdot \overline{\mathbb{P}}_{n,\ell}[\|x\|]$. The term $\overline{\mathbb{P}}_{n,\ell}[\|x\|] = \frac{1}{|\mathcal{C}_\ell|} \sum_{i \in \mathcal{C}_\ell} \mathbb{E}[\|\xi_i^{(k)}\|]$. From the proof of Theorem 2 (specifically Part 1c, relying on uniform integrability of moments of $\overline{\mathbb{P}}_n$), $\sup_n \mathbb{E}[\|\xi_i^{(k)}\|]$ is bounded for all $i$. Thus, $\sup_n \overline{\mathbb{P}}_{n,\ell}[\|x\|]$ is bounded (as $|\mathcal{C}_\ell| \to \infty$). By Assumption 3, $\left| \frac{|\mathcal{C}_\ell|}{n} - \pi_\ell \right| \to 0$. Therefore, $\mathbb{E}[T_{1,n}] \to 0$.

For $T_{2,n}$: We need to show $\mathbb{E}[W_1(\mathbb{P}_{n,\ell}, \mathbb{G}_\ell)] \to 0$ for each $\ell$. By the triangle inequality, $W_1(\mathbb{P}_{n,\ell}, \mathbb{G}_\ell) \leq W_1(\mathbb{P}_{n,\ell}, \overline{\mathbb{P}}_{n,\ell}) + W_1(\overline{\mathbb{P}}_{n,\ell}, \mathbb{G}_\ell)$. For the two terms on we have:

(a) $\mathbb{E}[W_1(\mathbb{P}_{n,\ell}, \overline{\mathbb{P}}_{n,\ell})] \to 0$: $\mathbb{P}_{n,\ell}$ is an empirical measure of $N_\ell = |\mathcal{C}_\ell|$ variables $\{\xi_i^{(k)} : i \in \mathcal{C}_\ell\}$. Since $N_\ell \to \infty$ (as $\pi_\ell > 0$), we can apply Proposition 8 to this specific subset of variables. The conditions for Proposition 8 are: (i) Uniform $\Psi_{r_{N_\ell}}$ sub-Gaussianity of $\langle \xi_i^{(k)}, \theta \rangle$ for $i \in \mathcal{C}_\ell$: This holds from Lemma 1. (ii) Variance of their empirical moments $\text{Var}(N_\ell^{-1} \sum_{i \in \mathcal{C}_\ell} \langle \xi_i^{(k)}, \theta \rangle^r) \to 0$ holds from the more general formulation of Lemma 6 where $\text{Cov}(\Delta_{i,\theta}^r, \Delta_{\theta,i'}^r) \lesssim n^{-1\{i \neq i'\}}$. (iii) $\sup_n M_1(\overline{\mathbb{P}}_{n,\ell}) < \infty$: This holds as shown for $T_{1,n}$. Thus, $\mathbb{E}[W_1(\mathbb{P}_{n,\ell}, \overline{\mathbb{P}}_{n,\ell})] \to 0$.

(b) $W_1(\overline{\mathbb{P}}_{n,\ell}, \mathbb{G}_\ell) \to 0$: We analyze the moments of $\overline{\mathbb{P}}_{n,\ell}$ for a given $\theta \in S^{d-1}$. $m_r(\overline{\mathbb{P}}_{n,\ell,\theta}) = \frac{1}{|\mathcal{C}_\ell|} \sum_{i \in \mathcal{C}_\ell} \mathbb{E}[\langle \xi_i^{(k)}, \theta \rangle^r]$. From Steps 1, 2, 3 of the proof of Proposition 4, we know that $\mathbb{E}[\langle \xi_i^{(k)}, \theta \rangle^r] = \mathbb{E}[\Delta_{i,\theta}^r] + o(1)$. If $r$ is odd, $\mathbb{E}[\Delta_{i,\theta}^r] = o(1)$ by Lemma 8. So $m_r(\overline{\mathbb{P}}_{n,\ell,\theta}) \to 0 = m_r(N(0, \theta^T \Sigma_\ell \theta))$. If $r$ is even, $\mathbb{E}[\Delta_{i,\theta}^r] = (r-1)!! \cdot \widetilde{\sigma}_{i,\theta}^r + o(1)$, where the $o(1)$ is uniform in $i$. From Step 4 in the proof of Proposition 4, for any $i \in \mathcal{C}_\ell$, $\widetilde{\sigma}_{i,\theta}^2 \to \sigma_{\ell,\theta}^{*2} := \theta^T \Sigma_\ell \theta$. Thus, for $i \in \mathcal{C}_\ell$, $\mathbb{E}[\langle \xi_i^{(k)}, \theta \rangle^r] \to (r-1)!! (\sigma_{\ell,\theta}^{*2})^{r/2} \cdot \mathbb{1}\{r \text{ is even}\}$. This limit is uniform for all $i \in \mathcal{C}_\ell$. Therefore,

$$m_r(\overline{\mathbb{P}}_{n,\ell,\theta}) = \frac{1}{|\mathcal{C}_\ell|} \sum_{i \in \mathcal{C}_\ell} \left( (r-1)!! (\sigma_{\ell,\theta}^{*2})^{r/2} \cdot \mathbb{1}\{r \text{ is even}\} + o(1) \right)$$

$$\to (r-1)!! (\sigma_{\ell,\theta}^{*2})^{r/2} \cdot \mathbb{1}\{r \text{ is even}\}.$$

This is $m_r(N(0, \theta^T \Sigma_\ell \theta))$. Since $\mathbb{G}_{\ell,\theta} = N(0, \theta^T \Sigma_\ell \theta)$ is determined by its moments, and its moments are finite, $\overline{\mathbb{P}}_{n,\ell,\theta} \rightsquigarrow \mathbb{G}_{\ell,\theta}$. The uniform integrability of moments for $\overline{\mathbb{P}}_{n,\ell,\theta}$ is inherited from the global case (as seen in $T_{1,n}$ argument, $M_p(\overline{\mathbb{P}}_{n,\ell})$ is bounded for any $p$). This promotes weak convergence to $W_1(\overline{\mathbb{P}}_{n,\ell,\theta}, \mathbb{G}_{\ell,\theta}) \to 0$. Then, by Proposition 8, $W_1(\overline{\mathbb{P}}_{n,\ell}, \mathbb{G}_\ell) \to 0$.

Since $\mathbb{E}[W_1(\mathbb{P}_{n,\ell}, \mathbb{G}_\ell)] \to 0$ for each $\ell$, and $\pi_\ell$ are constants, $\mathbb{E}[T_{2,n}] \to 0$. Combining $\mathbb{E}[T_{1,n}] \to 0$ and $\mathbb{E}[T_{2,n}] \to 0$ completes the proof. $\qquad\square$

**Proposition 6** (Part of Theorem 1). *Consider the settings of Proposition 4. Let $\widetilde{\mathbb{G}}_{n,\ell} = N(\mu_\ell, \Sigma_\ell/\nu_n)$. Then for any $R > 0$:*

$$\mathbb{E}\left\{ \sup_{f_1,\dots,f_L \in \mathrm{Lip}(R)} \left| \frac{1}{n} \sum_{\ell=1}^{L} \sum_{i \in \mathcal{C}_\ell} f_\ell(\overline{\phi}_i^{(k)}) - \sum_{\ell=1}^{L} \pi_\ell \mathbb{E}_{Y \sim \widetilde{\mathbb{G}}_{n,\ell}}[f_\ell(Y)] \right| \right\} \to 0.$$

*Proof.* Let $\widetilde{\mathbb{P}}_{n,\ell}$ be the class-conditional empirical measure for $\overline{\phi}_i^{(k)}$ for class $\ell$:

$$\widetilde{\mathbb{P}}_{n,\ell}[f] = \frac{1}{|\mathcal{C}_\ell|} \sum_{i \in \mathcal{C}_\ell} f(\overline{\phi}_i^{(k)}).$$

Let $\Delta'_n$ be the term inside the overall expectation:

$$\Delta'_n := \sup_{f_1,\dots,f_L \in \mathrm{Lip}(R)} \left| \sum_{\ell=1}^{L} \frac{|\mathcal{C}_\ell|}{n} \widetilde{\mathbb{P}}_{n,\ell}[f_\ell] - \sum_{\ell=1}^{L} \pi_\ell \widetilde{\mathbb{G}}_{n,\ell}[f_\ell] \right|.$$

Similar to the proof of Proposition 5, using the triangle inequality:

$$\Delta'_n \leq \sum_{\ell=1}^{L} \left| \frac{|\mathcal{C}_\ell|}{n} - \pi_\ell \right| \sup_{f_\ell \in \mathrm{Lip}(R)} |\widetilde{\mathbb{P}}_{n,\ell}[f_\ell]| \quad (:= T'_{1,n})$$

$$+ \sum_{\ell=1}^{L} \pi_\ell \sup_{f_\ell \in \mathrm{Lip}(R)} \left| \widetilde{\mathbb{P}}_{n,\ell}[f_\ell] - \widetilde{\mathbb{G}}_{n,\ell}[f_\ell] \right| \quad (:= T'_{2,n}).$$

The second term is $\sum_{\ell=1}^{L} \pi_\ell R \cdot W_1(\widetilde{\mathbb{P}}_{n,\ell}, \widetilde{\mathbb{G}}_{n,\ell})$. We can assume $f_\ell(0) = 0$ by replacing $f_\ell(x)$ with $f_\ell(x) - f_\ell(0)$ and noting that $|\widetilde{\mathbb{P}}_{n,\ell}[f_\ell(0)] - \widetilde{\mathbb{G}}_{n,\ell}[f_\ell(0)]| = |f_\ell(0) - f_\ell(0)| = 0$.

Before bounding the two terms, we first show that

$$\mathbb{E}\|\overline{\phi}_i^{(k)} - \mu_\ell\|_2 \to 0 \quad \text{uniformly for } i \in \mathcal{C}_\ell. \tag{25}$$

By Lemma 3 and Lemma 4, $\mathbb{E}[\overline{\phi}_i^{(k)}] \to \mu_\ell$ for $i \in \mathcal{C}_\ell$. Next,

$$\mathrm{Var}(\overline{\phi}_i^{(k)}) = \mathrm{Var}(\xi_i^{(k)}/\sqrt{\nu_n}) = \Sigma_\ell/\nu_n + o(\nu_n^{-1}),$$

uniformly over $i \in \mathcal{C}_\ell$, by noting that the convergence in the proof of Proposition 4 is, in fact, uniform over $i \in \mathcal{C}_\ell$ and $\theta \in \mathbb{S}^{d-1}$. Since $\mathbb{E}\|\overline{\phi}_i^{(k)} - \mu_\ell\|_2 \leq \mathbb{E}\|\overline{\phi}_i^{(k)} - \mathbb{E}[\overline{\phi}_i^{(k)}]\|_2 + \|\mathbb{E}[\overline{\phi}_i^{(k)}] - \mu_\ell\|_2$, and the first term is bounded by $\left(\mathrm{tr}(\mathrm{Var}(\overline{\phi}_i^{(k)}))\right)^{1/2} = O(\nu_n^{-1/2}) = o(1)$, and the second terms is $o(1)$ for $i \in \mathcal{C}_\ell$, the claim follows.

For $T'_{1,n}$: $\mathbb{E}[\sup_{f_\ell \in \mathrm{Lip}(R)} |\widetilde{\mathbb{P}}_{n,\ell}[f_\ell]|] \leq R \cdot \mathbb{E}[\widetilde{\mathbb{P}}_{n,\ell}[\|x\|]] = R \cdot \frac{1}{|\mathcal{C}_\ell|} \sum_{i \in \mathcal{C}_\ell} \mathbb{E}[\|\overline{\phi}_i^{(k)}\|]$. By eq. (25), $\mathbb{E}[\|\overline{\phi}_i^{(k)}\|]$ converges to $\|\mu_\ell\|$ which is bounded. Thus, $\sup_n \mathbb{E}[\sup_{f_\ell} |\widetilde{\mathbb{P}}_{n,\ell}[f_\ell]|]$ is bounded. Since $\left| \frac{|\mathcal{C}_\ell|}{n} - \pi_\ell \right| \to 0$ by Assumption 3, $\mathbb{E}[T'_{1,n}] \to 0$.

For $T'_{2,n}$: We need to show $\mathbb{E}[W_1(\widetilde{\mathbb{P}}_{n,\ell}, \widetilde{\mathbb{G}}_{n,\ell})] \to 0$ for each $\ell$. Let $f \in \mathrm{Lip}(R)$ with $f(0) = 0$. Let $\overline{\widetilde{\mathbb{P}}}_{n,\ell} = \mathbb{E}[\widetilde{\mathbb{P}}_{n,\ell}]$. We first analyze

$$|\overline{\widetilde{\mathbb{P}}}_{n,\ell}[f] - \widetilde{\mathbb{G}}_{n,\ell}[f]| \leq \frac{1}{|\mathcal{C}_\ell|} \sum_{i \in \mathcal{C}_\ell} \mathbb{E}|f(\overline{\phi}_i^{(k)}) - \widetilde{\mathbb{G}}_{n,\ell}[f]|,$$

where $\overline{\overline{\mathbb{P}}}_{n,\ell}[f] = \frac{1}{|\mathcal{C}_\ell|}\sum_{i\in\mathcal{C}_\ell}\mathbb{E}[f(\overline{\phi}_i^{(k)})]$. Using the decomposition for a single $i \in \mathcal{C}_\ell$:

$$\mathbb{E}|f(\overline{\phi}_i^{(k)}) - \widetilde{\mathbb{G}}_{n,\ell}[f]| \leq \mathbb{E}|f(\overline{\phi}_i^{(k)}) - f(\mu_\ell)| + \mathbb{E}|f(\mu_\ell) - \widetilde{\mathbb{G}}_{n,\ell}[f]|.$$

Let these two terms be $A_i, B_i$. (Note $B_i$ is actually independent of $i$ for $i \in \mathcal{C}_\ell$). Since $f \in \mathrm{Lip}(R)$:

- $A_i \leq R \cdot \mathbb{E}\|\overline{\phi}_i^{(k)} - \mu_\ell\|_2 = O(\nu_n^{-1/2})$ uniformly for $i \in \mathcal{C}_\ell$, by eq. (25).

- For $B_i$, we have

$$B_i = |\mathbb{E}_{Y\sim N(0,\Sigma_\ell/\nu_n)}[f(\mu_\ell) - f(\mu_\ell + Y)]| \leq R \cdot \mathbb{E}_{Y\sim N(0,\Sigma_\ell/\nu_n)}[\|Y\|_2]$$

and $\mathbb{E}_{Y\sim N(0,\Sigma_\ell/\nu_n)}[\|Y\|_2] \leq \sqrt{\mathrm{tr}(\Sigma_\ell/\nu_n)} = O(\nu_n^{-1/2})$. So $B_i = O(\nu_n^{-1/2})$.

Thus, uniformly over $i \in \mathcal{C}_\ell$ and $f \in \mathrm{Lip}(R)$, we have $\mathbb{E}|f(\overline{\phi}_i^{(k)}) - \widetilde{\mathbb{G}}_{n,\ell}[f]| = O(\nu_n^{-1/2})$. This establishes $W_1(\overline{\overline{\mathbb{P}}}_{n,\ell}, \widetilde{\mathbb{G}}_{n,\ell}) \to 0$.

Now, for the concentration part $\mathbb{E}[W_1(\widetilde{\mathbb{P}}_{n,\ell}, \overline{\overline{\mathbb{P}}}_{n,\ell})] \to 0$: We will verify the conditions of Proposition 8 for the variables $X_{i,n} = \overline{\phi}_i^{(k)}$ for $i \in \mathcal{C}_\ell$:

(i) Uniform $\Psi_{r_n}$ sub-Gaussianity of $\langle\overline{\phi}_i^{(k)},\theta\rangle$: Since $\overline{\phi}_i^{(k)} = \mathbb{E}[\overline{\phi}_i^{(k)}] + \xi_i^{(k)}/\sqrt{\nu_n}$, we have

$$\|\langle\overline{\phi}_i^{(k)},\theta\rangle\|_{\Psi_{r_n}} \leq \|\langle\mathbb{E}[\overline{\phi}_i^{(k)}],\theta\rangle\|_{\Psi_{r_n}} + \|\langle\xi_i^{(k)}/\sqrt{\nu_n},\theta\rangle\|_{\Psi_{r_n}}$$
$$= |\langle\mathbb{E}[\overline{\phi}_i^{(k)}],\theta\rangle| \cdot \|1\|_{\Psi_{r_n}} + \|\langle\xi_i^{(k)}/\sqrt{\nu_n},\theta\rangle\|_{\Psi_{r_n}}$$

the first term is bounded in the limit by $C\langle\mu_\ell,\theta\rangle$ where $C = \limsup_{n\to\infty}\|1\|_{\Psi_{r_n}}$ is a universal constant, and the second term is $O(\nu_n^{-1/2})$ by Lemma 1, both uniformly over $i \in \mathcal{C}_\ell$ and $\theta \in \mathbb{S}^{d-1}$.

and $\mu_\ell$ is bounded, and $\xi_i^{(k)}/\sqrt{\nu_n}$ has vanishing $\Psi$ norm (as $\xi_i^{(k)}$ has bounded $\Psi$ norm), $\langle\overline{\phi}_i^{(k)},\theta\rangle$ will have bounded $\Psi_{r_{N_\ell}}$ norm (dominated by $\langle\mu_\ell,\theta\rangle$ plus a small term).

(ii) Variance of empirical moments: $\mathrm{Var}(N_\ell^{-1}\sum_{i\in\mathcal{C}_\ell}\langle\overline{\phi}_i^{(k)},\theta\rangle^r)$. Again, we use $\overline{\phi}_i^{(k)} = \mathbb{E}[\overline{\phi}_i^{(k)}] + \xi_i^{(k)}/\sqrt{\nu_n}$. By an argument similar to Lemma 5, we obtain

$$\|\langle\overline{\phi}_i^{(k)},\theta\rangle^r - \langle\mathbb{E}\overline{\phi}_i^{(k)},\theta\rangle^r\|_{L^2} \leq \sum_{s=1}^r \binom{r}{s}\|\langle\mathbb{E}\overline{\phi}_i^{(k)},\theta\rangle\|_{L^{2r}}^{r-s} \cdot \|\langle\xi_i^{(k)}/\sqrt{\nu_n},\theta\rangle\|_{L^{2r}}^s \quad (26)$$

We have $\|\langle\mathbb{E}\overline{\phi}_i^{(k)},\theta\rangle\|_{L^{2r}}^{r-s} = |\langle\mathbb{E}\overline{\phi}_i^{(k)},\theta\rangle|^{r-s}$ since the quantity is deterministic. This is uniformly bounded over $i \in \mathcal{C}_\ell$ and $\theta \in \mathbb{S}^{d-1}$, by eq. (25). Similarly, $\|\langle\xi_i^{(k)},\theta\rangle\|_{L^{2r}}$ is uniformly bounded over $i \in \mathcal{C}_\ell$ and $\theta \in \mathbb{S}^{d-1}$, by the argument in the proof of Proposition 4 (the convergence of the moments is uniform over $i \in \mathcal{C}_\ell$). It follows that $\|\langle\xi_i^{(k)}/\sqrt{\nu_n},\theta\rangle\|_{L^{2r}}^s = O(\nu_n^{-s/2}) = O(\nu_n^{-1/2})$ for $s \geq 1$, uniformly over $i$ and $\theta$. The same then applies to LHS of eq. (26). This in turn implies $\|N_\ell^{-1}\sum_{i\in\mathcal{C}_\ell}\langle\overline{\phi}_i^{(k)},\theta\rangle^r - N_\ell^{-1}\sum_{i\in\mathcal{C}_\ell}\langle\mathbb{E}\overline{\phi}_i^{(k)},\theta\rangle^r\|_{L^2} = o(1)$. Now, $\mathrm{Var}(N_\ell^{-1}\sum_{i\in\mathcal{C}_\ell}\langle\mathbb{E}\overline{\phi}_i^{(k)},\theta\rangle^r) = 0$ since this quantity is deterministic. This implies (see the inequality in the proof of Lemma 2) $\mathrm{Var}(N_\ell^{-1}\sum_{i\in\mathcal{C}_\ell}\langle\overline{\phi}_i^{(k)},\theta\rangle^r) = o(1)$ which is the desired result.

(iii) $\sup_n M_1(\overline{\overline{\mathbb{P}}}_{n,\ell}) < \infty$: This was shown for $T'_{1,n}$.

Thus, by Proposition 8, $\mathbb{E}[W_1(\widetilde{\mathbb{P}}_{n,\ell}, \overline{\overline{\mathbb{P}}}_{n,\ell})] \to 0$.

Since $\mathbb{E}[W_1(\widetilde{\mathbb{P}}_{n,\ell}, \widetilde{\mathbb{G}}_{n,\ell})] \to 0$ for each $\ell$, it follows that $\mathbb{E}[T'_{2,n}] \to 0$. Combining $\mathbb{E}[T'_{1,n}] \to 0$ and $\mathbb{E}[T'_{2,n}] \to 0$ completes the proof. $\square$

# B Moment Characterization in $W_p$

In the following $\{\mathbb{H}_n\}_{n \geq 1}$ and $\mathbb{H}$ are all (Borel) probability measures on $\mathbb{R}^d$.

**Proposition 7.** *Assume that $\{\mathbb{H}_n\}_{n \geq 1}$ is a sequence of (Borel) measures on $\mathbb{R}^d$ such that*

$$\sup_{n \geq 1} \int |\theta^T x|^r d\mathbb{H}_n(x) < \infty, \quad \text{for all } \theta \in \mathbb{R}^d.$$

*Then, $\sup_{n \geq 1} \int \|x\|^r d\mathbb{H}_n(x) < \infty$.*

*Proof.* Let $\{\theta_1, \ldots, \theta_m\}$ be a $\frac{1}{2}$-net of the unit sphere $S^{d-1} = \{\theta \in \mathbb{R}^d : \|\theta\| = 1\}$. We have $\|x\| = \sup_{\theta \in S^{d-1}} |\theta^T x| \leq 2 \max_{i \in [m]} |\theta_i^T x|$. It follows that $\|x\|^r \leq 2^r \max_{i \in [m]} |\theta_i^T x|^r \leq 2^r \sum_{i=1}^m |\theta_i^T x|^r$, hence

$$\sup_{n \geq 1} \int \|x\|^r d\mathbb{H}_n(x) \leq 2^r \sum_{i=1}^m \sup_{n \geq 1} \int |\theta_i^T x|^r d\mathbb{H}_n(x) < \infty$$

proving the result. $\qquad\square$

# C $\Psi_r$ sub-Gaussians

**Definition 1** ($\Psi_r$ sub-Gaussian). Let $r \geq 2$ be a real number, and $\Psi_r : [0, \infty) \to [0, \infty)$ be defined by

$$\Psi_r(x) = \sum_{j=1}^{\lfloor r/2 \rfloor} \frac{x^{2j}}{j!}. \tag{27}$$

The corresponding Orlicz (or Luxembourg) norm for a random variable $X$ is:

$$\|X\|_{\Psi_r} = \inf\{K > 0 : \mathbb{E}[\Psi_r(|X|/K)] \leq 1\}. \tag{28}$$

**Lemma 9** (Norm equivalence). *Let $X$ be a random variable and $r \geq 2$. The following holds:*

(a) *Norm implies moments: If $\|X\|_{\Psi_r} \leq K$ for some $K > 0$, then*

$$(\mathbb{E}|X|^p)^{1/p} \leq C_1 K \sqrt{p} \quad \text{for all } p \in [2, 2\lfloor r/2 \rfloor]$$

*where $C_1 > 0$ is a universal constant.*

(b) *Moments imply norm: If $(\mathbb{E}|X|^p)^{1/p} \leq C \sqrt{p}$ for some $C > 0$ and for all $p \in [2, r]$, then*

$$\|X\|_{\Psi_r} \leq C_2 C$$

*where $C_2 = 2\sqrt{e}$.*

*Proof.* **Part (a)** Assume $\|X\|_{\Psi_r} \leq K$. By definition, $\mathbb{E}[\Psi_r(|X|/K)] \leq 1$.

$$\mathbb{E}\left[ \sum_{j=1}^{\lfloor r/2 \rfloor} \frac{(|X|/K)^{2j}}{j!} \right] \leq 1$$

For any integer $j_0 \in [1, \lfloor r/2 \rfloor]$, let $p = 2j_0$. Since all terms in the sum are non-negative:

$$\mathbb{E}\left[ \frac{|X|^p}{K^p j_0!} \right] \leq \mathbb{E}[\Psi_r(|X|/K)] \leq 1$$

So, $\mathbb{E}|X|^p \leq K^p j_0! = K^p (p/2)!$. Taking the $p$-th root: $(\mathbb{E}|X|^p)^{1/p} \leq K((p/2)!)^{1/p}$. Using the inequality $m! \leq e\sqrt{m}(m/e)^m$ for $m = p/2 \geq 1$:

$$((p/2)!)^{1/p} \leq (e\sqrt{p/2}(p/2e)^{p/2})^{1/p} = (e\sqrt{p/2})^{1/p}(p/2e)^{1/2} = (e\sqrt{p/2})^{1/p}\sqrt{\frac{p}{2e}}$$

The term $(e\sqrt{p/2})^{1/p}$ is bounded by a universal constant $c'$ for $p \geq 2$. (It tends to 1 as $p \to \infty$). Thus, $(\mathbb{E}|X|^p)^{1/p} \leq Kc'\sqrt{1/(2e)}\sqrt{p}$ for even integers $p \in [2, 2\lfloor r/2\rfloor]$. Now, let $p \in [2, 2\lfloor r/2\rfloor]$ be any real number. Let $q = 2\lceil p/2\rceil$. Then $q$ is an even integer, $p \leq q \leq p+1 < p+2$, and $q \leq 2\lceil(2\lfloor r/2\rfloor)/2\rceil = 2\lfloor r/2\rfloor$. By Lyapunov's inequality:

$$(\mathbb{E}|X|^p)^{1/p} \leq (\mathbb{E}|X|^q)^{1/q} \leq Kc'\sqrt{1/(2e)}\sqrt{q}$$

Since $q \leq p+2$ and $p \geq 2$, we have $q \leq p+p = 2p$. So $\sqrt{q} \leq \sqrt{2p} = \sqrt{2}\sqrt{p}$. Therefore, $(\mathbb{E}|X|^p)^{1/p} \leq Kc'\sqrt{1/(2e)}\sqrt{2}\sqrt{p} = (c'\sqrt{1/e})K\sqrt{p}$. Setting $C_1 = c'\sqrt{1/e}$ (a universal constant) proves the first part.

**Part (b)** Assume $(\mathbb{E}|X|^p)^{1/p} \leq C\sqrt{p}$ for $p \in [2, r]$. We want to find $k$ such that $\mathbb{E}[\Psi_r(|X|/k)] \leq 1$.

$$\mathbb{E}[\Psi_r(|X|/k)] = \sum_{j=1}^{\lfloor r/2\rfloor} \frac{\mathbb{E}[|X|^{2j}]}{k^{2j}j!}$$

Let $p = 2j$. Since $j \in [1, \lfloor r/2\rfloor]$, $p \in [2, 2\lfloor r/2\rfloor]$. This range is contained in $[2, r]$. So we can use the moment bound: $\mathbb{E}|X|^p \leq (C\sqrt{p})^p = C^p p^{p/2}$.

$$\mathbb{E}[\Psi_r(|X|/k)] \leq \sum_{j=1}^{\lfloor r/2\rfloor} \frac{C^{2j}(2j)^j}{k^{2j}j!}$$

Using the bound $(2j)^j/j! \leq (2e)^j$:

$$\mathbb{E}[\Psi_r(|X|/k)] \leq \sum_{j=1}^{\lfloor r/2\rfloor} \frac{C^{2j}(2e)^j}{k^{2j}} = \sum_{j=1}^{\lfloor r/2\rfloor} \left(\frac{2eC^2}{k^2}\right)^j$$

This is a geometric series with ratio $R = 2eC^2/k^2$. If we choose $k$ such that $R \leq 1/2$, the sum is bounded by $\sum_{j=1}^{\infty}(1/2)^j = 1$. We need $2eC^2/k^2 \leq 1/2$, which means $k^2 \geq 4eC^2$. Let $k = \sqrt{4e}C = 2\sqrt{e}C$. With this choice of $k$, we have $\mathbb{E}[\Psi_r(|X|/k)] \leq 1$. By the definition of the norm, $\|X\|_{\Psi_r} \leq k = 2\sqrt{e}C$. Setting $C_2 = 2\sqrt{e}$ proves the second part. $\qquad\square$

**Lemma 10** (Tail bound). *Let $Y$ be a random variable and $r \geq 2$. Suppose $\|Y\|_{\Psi_r} \leq K$ for some $K > 0$. Then there exists a universal constant $c_0 > 0$ such that for all $t \geq c_0 K$:*

$$\mathbb{P}(|Y| \geq t) \leq \exp\left(-c_1 \min\left\{\frac{t^2}{K^2}, \lfloor r/2\rfloor\right\}\right)$$

*where $c_1 = 1/(4C_1^2 e)$ and $C_1$ is the universal constant from Lemma 9(a). The threshold constant is $c_0 = 2C_1\sqrt{e}$.*

*Proof.* The assumption $\|Y\|_{\Psi_r} \leq K$ implies $(\mathbb{E}|Y|^p)^{1/p} \leq C_1 K\sqrt{p}$ for all $p \in [2, 2\lfloor r/2\rfloor]$ by Lemma 9(a). Let $r'_0 = 2\lfloor r/2\rfloor$. This matches the condition (56) of [18, Lemma 25] with $\Delta = Y$, $\eta = 1/2$, $K_{lem} = K$, $C_{lem} = 2C_1^2$, and $r_0$ replaced by $r'_0$. Lemma 25 applies for $x \geq 4C_{lem}\eta e = 4(2C_1^2)(1/2)e = 4C_1^2 e$. It gives the tail bound:

$$\mathbb{P}(|Y| \geq Kx^{1/2}) \leq \exp\left(-\min\left\{\frac{x}{2C_{lem}e}, \eta r'_0\right\}\right) = \exp\left(-\min\left\{\frac{x}{4C_1^2 e}, \lfloor r/2\rfloor\right\}\right)$$

Let $t = Kx^{1/2}$, so $x = (t/K)^2$. The condition on $x$ becomes $t \geq K\sqrt{4C_1^2 e} = 2C_1\sqrt{e}K$. Substituting $x$ in the bound yields:

$$\mathbb{P}(|Y| \geq t) \leq \exp\left(-\min\left\{\frac{(t/K)^2}{4C_1^2 e}, \lfloor r/2\rfloor\right\}\right)$$

Setting $c_1 = 1/(4C_1^2 e)$ and $c_0 = 2C_1\sqrt{e}$ gives the desired result. $\qquad\square$

# D  $W_1$ Concentration of the Empirical Measure

We use the following general concentration result for the empirical measure of dependent variables, established in the companion paper Amini and Vinas (2026) [2]. We reproduce the result here for reference.

**Proposition 8.** *Let $\mu_n = \frac{1}{n} \sum_{i=1}^n \delta_{Y_{i,n}}$ be the empirical measure of vector-valued random variables $Y_{i,n} \in \mathbb{R}^d$ for $i \in [n]$, and let $\bar{\mu}_n = \mathbb{E}\mu_n$. Assume that for some sequence $r_n = \omega(1)$ and for any $\theta \in S^{d-1}$,*

*(a)* $\{\langle \theta, Y_{i,n} \rangle\}_{i=1}^n$ *is uniformly $\Psi_{r_n}$ sub-Gaussian, that is, there exists $\zeta(\theta) > 0$, such that $\sup_{i \in [n]} \|\langle \theta, Y_{i,n} \rangle\|_{\Psi_{r_n}} \le \zeta(\theta)$.*

*(b)* $\mathrm{var}\big(n^{-1} \sum_{i=1}^n \langle \theta, Y_{i,n} \rangle^r\big) \to 0$ *as $n \to \infty$ for all $r \in \mathbb{N}$.*

*(c)* $\sup_{n \ge 1} M_1(\bar{\mu}_n) < \infty$.

*Then, $\mathbb{E}\big[W_1(\mu_n, \bar{\mu}_n)\big] \to 0$ as $n \to \infty$.*

The proof of Proposition 8 relies on approximating Lipschitz test functions with polynomials via Jackson kernels. This approach requires precise control over the growth of the polynomial coefficients. For completeness, we list the key approximation lemmas used in [2] to establish the result.

**Lemma 11.** *Let $T_k$ be the $k$th Chebyshev polynomial, and let $[T_k]_j$ be the coefficient of $x^j$ in $T_k(x)$. Then, $|[T_k]_0| \le 1$ and*

$$\max_{1 \le j \le k} |[T_k]_j| \le (1 + \sqrt{2})^k \le 3^k.$$

**Lemma 12** (Chebyshev–Jackson approximation). *Let $B \ge 3$. Then, for any $f : [-B, B] \to \mathbb{R}$ 1-Lipschitz with $f(0) = 0$, there exists a polynomial $P(x) = \sum_{j=0}^m c_j x^j$, with $m \in 4\mathbb{N}$, such that*

$$\sup_{x \in [-B,B]} |f(x) - P(x)| \le \frac{18B}{m}, \quad |c_j| \le 6B \cdot 3^{m-j}, \quad \text{for all } j \ge 0.$$

We refer the reader to [2] for the detailed proofs of Proposition 8 and the derivation of the coefficient bounds.

# E  Remaining proofs

## E.1  Proof of Lemma 6

Let $\mathcal{W}_k(i)$ be the set of directed, length $k$ walks starting at node $i \in [n]$. We consider $r$-tuples of walks called *walk sequences* where $\boldsymbol{w} \in \mathcal{W}_k^r(i)$ gives $\boldsymbol{w} = (\boldsymbol{w}^s)_{s=1}^r$ with $\boldsymbol{w}^s \in \mathcal{W}_k(i)$. We define the last vertex projection $\mathfrak{p} : \mathcal{W}_k(i) \to [n]$ and walk products $A_{\boldsymbol{w}^s} := \prod_{\ell=1}^k A_{i_\ell j_\ell}$ with $\boldsymbol{w}^s = ((i_\ell, j_\ell))_{\ell=1}^k$.

Relating back to $\Delta_{i,\theta}$, let

$$\varrho(\boldsymbol{w}) = \mathbb{E}\Big[ \prod_{s=1}^r (A_{\boldsymbol{w}^s} - \mathbb{E}[A_{\boldsymbol{w}^s}]) x_{\mathfrak{p}(\boldsymbol{w}^s)} \Big]$$

with $x := X\theta$. Then

$$\mathbb{E}[\mathring{\Delta}_{i,\theta}^r] = \sum_{\boldsymbol{w} \in \mathcal{W}_k^r(i)} \varrho(\boldsymbol{w}).$$

Further let $[w]$ and $[\![w]\!]$ be the set of unique edges and vertices, respectively, found on a walk $w$. A walk sequence $\boldsymbol{w}$ is said to be *overlapping* if for every $s \in [r]$ there exists a distinct $s' \in [r]$ such that $[\boldsymbol{w}^s] \cap [\boldsymbol{w}^{s'}] \ne \varnothing$. Walk sequence which are not overlapping have $\varrho(\boldsymbol{w}) = 0$. For this reason we define the following walk sets

$$\mathcal{N}_{r,t,v}(i) := \{\boldsymbol{w} \in \mathcal{W}_k^r(i) : \boldsymbol{w} \text{ overlapping}, \ |[\boldsymbol{w}]| = t, \ |[\![\boldsymbol{w}]\!]| = v\} \tag{29}$$

where $[\boldsymbol{w}] := \bigcup_{s=1}^r [\boldsymbol{w}^s]$ and $[\![\boldsymbol{w}]\!] := \bigcup_{s=1}^r [\![\boldsymbol{w}^s]\!]$.

The walk sets $\{\mathcal{N}_{r,t,v}(i)\}_{t,v}$ form a partition for $\mathcal{W}_k^r(i)$ with $2 \leq v \leq t+1$ and $1 \leq t \leq t_*$ where $t_* \leq rk - \lceil r/2 \rceil$. This gives the sum equivalence

$$\sum_{\boldsymbol{w} \in \mathcal{W}_k^r(i)} \varrho(\boldsymbol{w}) = \sum_{t=1}^{t_*} \sum_{v=2}^{t+1} \sum_{\boldsymbol{w} \in \mathcal{N}_{r,t,v}(i)} \varrho(\boldsymbol{w}),$$

which gives fine-grained control of $\varrho(\boldsymbol{w})$ for the specific walk sets $\mathcal{N}_{r,t,v}(i)$.

To prove the result, start by expanding the variance of the $r$-empirical moment of $\gamma$,

$$\mathrm{Var}\Big(\frac{1}{n} \sum_{i=1}^n \Delta_{i,\theta}^r \Big) = \frac{1}{n^2} \sum_{i,i'} \mathbb{E}[\Delta_{i,\theta}^r \Delta_{i',\theta}^r] - \mathbb{E}[\Delta_{i,\theta}^r]\mathbb{E}[\Delta_{i',\theta}^r]. \tag{30}$$

By the $n^{-2}$ scaling over $i, i' \in [n]$, it suffices to show

$$\mathrm{Cov}\big(\Delta_{i,\theta}^r, \Delta_{\theta,i'}^r\big) = \mathbb{E}[\Delta_{i,\theta}^r \Delta_{i',\theta}^r] - \mathbb{E}[\Delta_{i,\theta}^r]\mathbb{E}[\Delta_{\theta,i'}^r] \lesssim n^{-1\{i \neq i'\}},$$

for every $i, i' \in [n]$.

Introduce the new notation for walk-sequence pairs $(\boldsymbol{w}, \widetilde{\boldsymbol{w}})$

$$\varrho(\boldsymbol{w}, \widetilde{\boldsymbol{w}}) = \mathbb{E}\Big\{ \Big( \prod_{s=1}^r (A_{\boldsymbol{w}^s} - \mathbb{E}[A_{\boldsymbol{w}^s}]) x_{\mathfrak{p}(\boldsymbol{w}^s)} \Big) \Big( \prod_{s=1}^r (A_{\widetilde{\boldsymbol{w}}^s} - \mathbb{E}[A_{\widetilde{\boldsymbol{w}}^s}]) x_{\mathfrak{p}(\widetilde{\boldsymbol{w}}^s)} \Big) \Big\}.$$

Then, the walk-linearized covariance expansion is

$$\mathrm{Cov}\big(\Delta_{i,\theta}^r, \Delta_{\theta,i'}^r\big) = \frac{1}{\nu_n^{r(2k-1)}} \sum_{(\boldsymbol{w}, \widetilde{\boldsymbol{w}}) \in \mathcal{W}_k^r(i) \times \mathcal{W}_k^r(i')} \varrho(\boldsymbol{w}, \widetilde{\boldsymbol{w}}) - \varrho(\boldsymbol{w})\varrho(\widetilde{\boldsymbol{w}}). \tag{31}$$

We are interested in the case $\varrho(\boldsymbol{w}, \widetilde{\boldsymbol{w}})$ does not factorize as $\varrho(\boldsymbol{w}, \widetilde{\boldsymbol{w}}) = \varrho(\boldsymbol{w})\varrho(\widetilde{\boldsymbol{w}})$. Collect walk pairs under the concatenation notation $\boldsymbol{w}|\widetilde{\boldsymbol{w}} = (\boldsymbol{w}^1, \ldots, \boldsymbol{w}^r, \widetilde{\boldsymbol{w}}^1, \ldots, \widetilde{\boldsymbol{w}}^r)$ and define the walk set

$$\mathcal{M}_{r,t,v} := \{(\boldsymbol{w}, \widetilde{\boldsymbol{w}}) \in \mathcal{W}_k^r(i) \times \mathcal{W}_k^r(i') : \boldsymbol{w}|\widetilde{\boldsymbol{w}} \text{ overlapping}, |[\![\boldsymbol{w}|\widetilde{\boldsymbol{w}}]\!]| = t, |[\![\boldsymbol{w}|\widetilde{\boldsymbol{w}}]\!]| = v, |[\boldsymbol{w}] \cap [\widetilde{\boldsymbol{w}}]| > 0\}. \tag{32}$$

The last condition of (32) filters out walk pairs $(\boldsymbol{w}, \widetilde{\boldsymbol{w}})$ which factorize as $\varrho(\boldsymbol{w}, \widetilde{\boldsymbol{w}}) = \varrho(\boldsymbol{w})\varrho(\widetilde{\boldsymbol{w}})$. Similarly, if $\boldsymbol{w}|\widetilde{\boldsymbol{w}}$ is not overlapping $\varrho(\boldsymbol{w}, \widetilde{\boldsymbol{w}}) = 0$ and, consequently, $\varrho(\boldsymbol{w})\varrho(\widetilde{\boldsymbol{w}}) = 0$.

Let's start with the case $i = i'$. By the set construction in (32), $\mathcal{M}_{r,t,v}(i, i) \subseteq \mathcal{N}_{2r,t,v}(i)$. So $|\mathcal{M}_{r,t,v}(i, i)| \leq |\mathcal{N}_{2r,t,v}(i)|$ and by the counting result [18, Lemma 13]

$$|\mathcal{M}_{r,t,v}(i, i)| \leq (v-1)^{2rk} \binom{n-1}{v-1}. \tag{33}$$

A similar argument can be made when $i$ and $i'$ are distinct. By fixing $i$ and $i'$, we are left selecting $\binom{n-2}{v-2}$ unique vertices with a walk selection factor of $(v-1)^{2rk}$. Altogether,

$$|\mathcal{M}_{r,t,v}(i, i')| \leq (v-1)^{2rk} \binom{n-2}{v-2}. \tag{34}$$

For bounds on $v$ and $t$, we note that $\boldsymbol{u} := \boldsymbol{w}|\widetilde{\boldsymbol{w}}$ is an overlapping walk sequence, which by the partition result [18, Lemma 12], means it must have, at most, $|[\![\boldsymbol{u}]\!]| \leq 2rk - r$ unique edges. Similarly, the number of unique vertices bounds as $|[\![\boldsymbol{u}]\!]| \leq |[\boldsymbol{u}]| + 1$ since the discrete graph $([\![\boldsymbol{u}]\!], [\boldsymbol{u}])$ associated with $\boldsymbol{u}$ is necessarily connected by the rooted nature of the walks in the sequence $\boldsymbol{u}$ (walks must start at $i$ or $i'$) and the last condition of (32).

Next, we consider the bound $|\varrho(\boldsymbol{w}, \widetilde{\boldsymbol{w}})| \leq 2 \max\{|\varrho(\boldsymbol{w}, \widetilde{\boldsymbol{w}})|, |\varrho(\boldsymbol{w})\varrho(\widetilde{\boldsymbol{w}})|\}$. Introduce the notation, $\varrho_1(\boldsymbol{w}) = \mathbb{E}\big[\prod_{s=1}^r (A_{\boldsymbol{w}^s} - \mathbb{E}[A_{\boldsymbol{w}^s}])\big]$ and $\varrho_2(\boldsymbol{w}) = \mathbb{E}\big[\prod_{s=1}^r x_{\mathfrak{p}(\boldsymbol{w}^s)}\big]$. We analogously define, $\varrho_1(\boldsymbol{w}, \widetilde{\boldsymbol{w}}) := \varrho_1(\boldsymbol{w}|\widetilde{\boldsymbol{w}})$ and $\varrho_2(\boldsymbol{w}, \widetilde{\boldsymbol{w}}) := \varrho_2(\boldsymbol{w}|\widetilde{\boldsymbol{w}})$. From [18, Lemma 10],

$$|\varrho_1(\boldsymbol{w})\varrho_1(\widetilde{\boldsymbol{w}})| \leq 2^{2r}(\nu_n/n)^{|[\boldsymbol{w}]| + |[\widetilde{\boldsymbol{w}}]|} \leq 2^{2r}(\nu_n/n)^{|[\boldsymbol{w}|\widetilde{\boldsymbol{w}}]|} \quad \text{and} \quad |\varrho_1(\boldsymbol{w}, \widetilde{\boldsymbol{w}})| \leq 2^{2r}(\nu_n/n)^{|[\boldsymbol{w}|\widetilde{\boldsymbol{w}}]|}$$

and
$$|\varrho_2(\boldsymbol{w})\varrho_2(\widetilde{\boldsymbol{w}})| \leq (2\sqrt{r}\kappa_0)^{2r} \quad \text{and} \quad |\varrho_2(\boldsymbol{w},\widetilde{\boldsymbol{w}})| \leq (2\sqrt{r}\kappa_0)^{2r}$$
where $\kappa_0$ is defined as in Proposition 3. Let $t_* = r(2k-1)$ then

$$\mathrm{Cov}\big(\Delta_{i,\theta}^r, \Delta_{\theta,i'}^r\big) \leq \frac{1}{\nu_n^{r(2k-1)}} \sum_{t=1}^{t_*} \sum_{v=2}^{t+1} (4\sqrt{r}\kappa_0)^{2r} \cdot |\mathcal{M}_{r,t,v}(i,i')|(\nu_n/n)^t. \tag{35}$$

For the case $i = i'$, cardinality and $|\mathcal{M}_{r,t,v}(i,i)| \lesssim n^{v-1} \leq n^t$ by (33).

$$\mathrm{Cov}\big(\Delta_{i,\theta}^r, \Delta_{\theta,i'}^r\big) \lesssim \frac{1}{\nu_n^{r(2k-1)}} \sum_{t=1}^{t_*} \sum_{v=2}^{t+1} (4\sqrt{r}\kappa_0)^{2r} \cdot \nu_n^t$$

$$\lesssim \frac{\nu_n^{t_*}}{\nu_n^{r(2k-1)}},$$

where the last line follows from the fact $r$ and $k$ are fixed relative to $n$. Similarly for the off-diagonal case of $i \neq i'$, $|\mathcal{M}_{r,t,v}(i,i')| \lesssim n^{v-2} \leq n^{t-1}$ by (34) and

$$\frac{1}{\nu_n^{r(2k-1)}} \sum_{t=1}^{t_*} \sum_{v=2}^{t+1} (4\sqrt{r}\kappa_0)^{2r} \cdot |\mathcal{M}_{r,t,v}(i,i')|(\nu_n/n)^t \lesssim \frac{1}{\nu_n^{r(2k-1)}} \cdot \frac{\nu_n^{t_*}}{n}.$$

Noting $t_* = r(2k-1)$, this proofs the claim that $\mathrm{Cov}\big(\Delta_{i,\theta}^r, \Delta_{\theta,i'}^r\big) \lesssim n^{-1\{i \neq i'\}}$.

### E.2 Proof of Lemma 7

Shown in [18] the dominant term in a walk-based for $\mathring{\Delta}_{i,\theta}$ is given by the proxy term

$$\widetilde{T}_i^{\mathrm{hi}}(r) = (r-1)!! \sum_{(j_\ell)_\ell \in \mathcal{P}_{[n]\setminus\{i\}}^{r/2}} \prod_{q=1}^{r/2} p_{ij_\ell}(1 - p_{ij_\ell})(e_{j_\ell}^T \mathbb{E}[A]^{k-1}\mathbb{E}[X]\theta)^2$$

where $\mathcal{P}_{[n]\setminus\{i\}}^{r/2}$ is the set of coordinate distinct $(r/2)$-tuples on $[n] \setminus \{i\}$. Specifically, it was shown for $r \in 2\mathbb{N}$ and $\nu_n$ sufficiently large

$$\big|\mathbb{E}[\Delta_{i,\theta}^r] - \nu_n^{-(rk-r/2)} \widetilde{T}_i^{\mathrm{hi}}(r)\big| \leq C(r)x_*^r(n^{-1} + \nu_n^{-\epsilon}) \tag{36}$$

where $\epsilon$ can be used to parameterize the separation of higher- and lower-order terms $\mathring{\Delta}_{i,\theta}$ [18, Lemma 14 and Lemma 18].

To obtain the limiting closed form, we utilize $|[n]^{r/2} \setminus \mathcal{P}_{[n]\setminus\{i\}}^{r/2}| \leq C(r)n^{r/2-1}$ and

$$\sum_{(j_\ell)_\ell \in [n]^{r/2}} \prod_{q=1}^{r/2} p_{ij_\ell}(1 - p_{ij_\ell})(e_{j_\ell}^T \mathbb{E}[A]^{k-1}\mathbb{E}[X]\theta)^2 = \Big( \sum_{j \in [n]} p_{ij}(1 - p_{ij})(e_j^T \mathbb{E}[A]^{k-1}\mathbb{E}[X]\theta)^2 \Big)^{r/2}$$

$$= \big((\mathbb{E}[A]^{k-1}\mathbb{E}[X]\theta)^T (\nu_n V_i^2)(\mathbb{E}[A]^{k-1}\mathbb{E}[X]\theta)\big)^{r/2}.$$

For brevity, let $f_i(j) := (p_{ij}/\nu_n)(1 - p_{ij})(e_j^T \mathbb{E}[A/\nu_n]^{k-1}\mathbb{E}[X]\theta)^2$. Then, noting $\mathcal{P}_{[n]\setminus\{i\}}^{r/2} \subseteq [n]^{r/2}$,

$$|\nu_n^{-(rk-r/2)} \widetilde{T}_i^{\mathrm{hi}}(r) - (r-1)!! \|V_i\mathbb{E}[A]^{k-1}\mathbb{E}[X]\theta\|_2^r|$$

$$= (r-1)!! \Big| \sum_{(j_\ell)_\ell \in \mathcal{P}_{[n]\setminus\{i\}}^{r/2}} \prod_{q=1}^{r/2} f_i(j_\ell) - \sum_{(j_\ell)_\ell \in [n]^{r/2}} \prod_{q=1}^{r/2} f_i(j_\ell) \Big|$$

$$\leq (r-1)!! \, |[n]^{r/2} \setminus \mathcal{P}_{[n]\setminus\{i\}}^{r/2}| \, (\max_{j \in [n]} f_i(j))^{r/2}.$$

Let $\mathcal{W}_{k-1}(j)$ be the set of $k-1$ walks on $[n]$ starting at $j$. Then, with $\mathcal{W}_{k-1}^2(j) := \mathcal{W}_{k-1}(j) \times \mathcal{W}_{k-1}(j)$

$$f_i(j) = (p_{ij}/\nu_n)(1 - p_{ij}) \sum_{\boldsymbol{w} \in \mathcal{W}_{k-1}^2(j)} \prod_{s=1}^{2} \Big( \mathbb{E}[(X\theta)_{\mathfrak{p}(\boldsymbol{w}^s)}] \prod_{\ell=1}^{k-1} (p_{(\boldsymbol{w}^s)_\ell}/\nu_n) \Big)$$

Recall that $\mathbb{E}|(X\theta)_i| < x_*$ by assumption. Since $|\mathcal{W}_{k-1}(j)| \le |[n]^{k-1}| = n^{k-1}$ and $p_{ij}/\nu_n \le 1/n$ we have

$$f_i(j) \le x_*^2/n \qquad \text{for every } i, j \in [n].$$

Altogether, this yields the inequality

$$|\nu_n^{-(rk-r/2)}\widetilde{T}_i^{\text{hi}}(r) - (r-1)!! \, \|V_i \mathbb{E}[A]^{k-1}\mathbb{E}[X]\theta\|_2^r| \le C(r) x_*^r n^{-1},$$

where constants not depending on $r$ or $x_*$ have been absorbed in $C$. Noting that $n^{-1} \le \nu_n^{-\epsilon}$ for $0 < \epsilon < 1$ and piecing together with (36) produces the desired bound.

### E.3   Proof of Lemma 3

Similar to the proof Lemma 6 we begin with a walk analysis. Define the simple walk partition element $\mathcal{N}_{t,v}(i) \coloneqq \{w \in \mathcal{W}_k(i) : \, |[w]| = t, \, |[\![w]\!]| = v\}$. Note that, in this case, $\mathcal{N}_{t,v}(i)$ no longer has an overlapping constraint. As such,

$$\mathbb{E}[\langle \overline{\phi}_i^{(k)}, \theta\rangle] = \frac{1}{\nu_n^k} \sum_{t=1}^{k}\sum_{v=2}^{t+1} \sum_{w \in \mathcal{N}_{t,v}(i)} \mathbb{E}[A_w]\mathbb{E}[(X\theta)_{\mathfrak{p}(w)}]$$

where notation $A_w \coloneqq \prod_{\ell=1}^k A_{i_\ell j_\ell}$. Similarly define $(\mathbb{E}[A])_w = \prod_{\ell=1}^k \mathbb{E}[A_{i_\ell j_\ell}]$ for walks on the expected matrix $\mathbb{E}[A]$. Note that, when $t = v - 1 = k$, the edges of $w$ are all unique and the expectation factorizes as

$$\sum_{w \in \mathcal{N}_{k,k+1}(i)} \mathbb{E}[A_w]\mathbb{E}[(X\theta)_{\mathfrak{p}(w)}] = \sum_{w \in \mathcal{N}_{k,k+1}(i)} (\mathbb{E}[A])_w \, \mathbb{E}[(X\theta)_{\mathfrak{p}(w)}].$$

Therefore,

$$\mathbb{E}[\langle \overline{\phi}_i^{(k)}, \theta\rangle] - \langle \gamma_i^T, \theta\rangle = \frac{1}{\nu_n^k} \sum_{t=1}^{k}\sum_{v=2}^{t+1} \sum_{w \in \mathcal{N}_{t,v}(i)} (\mathbb{E}[A_w] - (\mathbb{E}[A])_w)\mathbb{E}[(X\theta)_{\mathfrak{p}(w)}] \cdot 1\{v \neq k+1\}.$$

Setting $r = 1$ in Lemma 13 of [18] gives the counting bound $\sum_{v=2}^{b+1} |\mathcal{N}_{t,v}(i)| \le b^{k-b}(en)^b$. Finally, noting that $\mathbb{E}[A_w] - (\mathbb{E}[A])_w \le 2(\nu_n/n)^{|w|}$,

$$\|\mathbb{E}[\langle \overline{\phi}_i^{(k)}, \theta\rangle] - \langle \gamma_i^T, \theta\rangle\|_2 \le \frac{1}{\nu_n^k} \sum_{t=1}^{k}\sum_{v=2}^{t+1} |\mathcal{N}_{t,v}(i)|(\nu_n/n)^r x_* \cdot 1\{v \neq k+1\}$$

$$= \frac{1}{\nu_n^k} \sum_{v=2}^{k} |\mathcal{N}_{t,v}(i)|(\nu_n/n)^r x_* + \frac{1}{\nu_n^k} \sum_{t=1}^{k-1}\sum_{v=2}^{t+1} |\mathcal{N}_{t,v}(i)|(\nu_n/n)^r x_*$$

$$\le \frac{1}{n}(k-1)^{k-1}e^{k-1} x_* + \frac{1}{\nu_n^k} \sum_{t=1}^{k-1} t^{k-t} e^t x_* \nu_n^t$$

$$\le C(k) x_* \nu_n^{-1}.$$

Since the above holds for any $i \in [n]$ and any $\theta \in \mathbb{S}^{d-1}$,

$$\max_{i \in [n]} \|\mathbb{E}[\overline{\phi}_i^{(k)}] - \gamma_i^T\| = \max_{i \in [n]} \max_{\theta \in \mathbb{S}^{d-1}} \|\langle(\mathbb{E}[\overline{\phi}_i^{(k)}] - \gamma_i^T), \theta\rangle\|_2 \le C(k) x_* \nu_n^{-1}.$$

## F   Joint Wasserstein Distance and the Class-Conditional Supremum

In Theorem 1, we state that the joint empirical distribution converges in 1-Wasserstein distance and then provide a related, stronger-looking class-conditional convergence statement (11). This note formalizes the relationship between these two quantities, showing that the latter is a tractable upper bound on the former.

Consider the joint space $[L] \times \mathbb{R}^d$ with the metric $d\big((z_1, y_1), (z_2, y_2)\big) \coloneqq \mathbb{1}\{z_1 \neq z_2\} + \|y_1 - y_2\|_2$. The true joint 1-Wasserstein distance is the supremum of the difference in expectations over all 1-Lipschitz functions $F : [L] \times \mathbb{R}^d \to \mathbb{R}$.

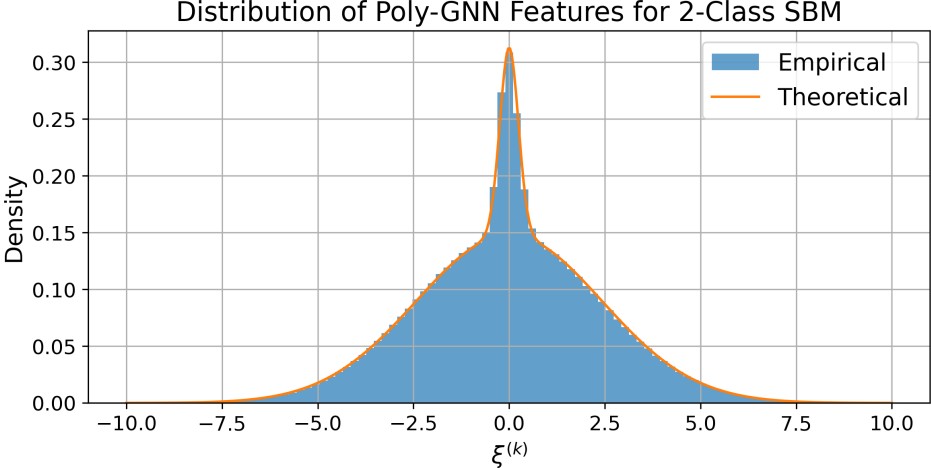

Figure 5: Empirical distribution of a two-class CSBM with exaggerated class proportions and edge probabilities. Both mixture components are centered at zero with a visible difference between the peak widths and heights of each component.

A function $F$ is 1-Lipschitz with respect to this metric if and only if its component functions, $f_\ell(y) := F(\ell, y)$, satisfy two conditions: (1) Each $f_\ell : \mathbb{R}^d \to \mathbb{R}$ is 1-Lipschitz. (2) The collection $\{f_\ell\}_{\ell=1}^L$ is jointly coupled by the constraint $|f_{\ell_1}(y_1) - f_{\ell_2}(y_2)| \leq 1 + \|y_1 - y_2\|_2$ for any $\ell_1 \neq \ell_2$.

In contrast, the class-conditional expression in Eq. (11) takes its supremum over all possible collections of 1-Lipschitz functions $\{f_\ell\}$ without enforcing the second joint constraint.

The set of test functions for the true joint $W_1$ distance is therefore a strict subset of the test functions for the class-conditional expression. Consequently, the class-conditional expression provides a valid upper bound on the joint 1-Wasserstein distance. This justifies our proof strategy: showing that this upper bound converges to zero is a sufficient condition to prove the desired joint convergence.

## G   Simulation Details for Figures

This appendix details the experimental setups for the figures presented in the main text. Specific parameters for these and all other figures are provided in the subsequent subsections.

### G.1   Details for Figure 1

The plots in Figure 1 were simulated from a 1-class SBM, commonly referred to as an Erdős–Réyni graph, with probability parameter $p = \nu_n/n$. Depth $k = 3$ was used with unit, univariate features $X_i = 1$ for all $i \in [n]$. A grid search was performed on graph sizes $n \in \{300, 3000, 30000\}$ with expected degrees $\nu_n \in \{2, 4, 16\}$. These graph are very sparse, yet they approach Gaussianity fairly quickly. Particularly, the plot associated with $\nu_n = 16$ has nearly symmetrical tails and a bell curve shape.

### G.2   Details for Figure 2

The plots in Figure 2 were generated using a 3-class CSBM with $n = 8192$ nodes. Class proportions were $\pi_1 = 0.25, \pi_2 = 0.45, \pi_3 = 0.30$, average degree parameter was $\nu_n = \sqrt{8192}$, and the inter-community probability scaling matrix was $B = (\nu_n/n) \cdot \begin{pmatrix} 0.4 & 1 & 1 \\ 1 & 0.4 & 1 \\ 1 & 1 & 0.4 \end{pmatrix}$. Initial features $X_i$ where $d = 2$ dimensional and generated as $X_i \sim N(M_{z_i,*}, \sigma^2 I_2)$ with $\sigma^2 = 0.25$ and $M_{1,*} = [2, 2]^T$, $M_{2,*} = [-1, -3]^T$, and $M_{3,*} = [-1, 0]^T$.

Cross entropy training was run for a single linear classifier layer for 10 epochs with learning rate 10 on the SGD optimization. Although small differences are expected at later time steps, Figure 2 still shows good agreement between the empirical and theoretical gradient average.

### G.3 Details for Figure 3

The plots in Figure 3 were generated using a 2-class SBM with $n = 32000$ nodes. Class proportions were $\pi_1 = 0.4, \pi_2 = 0.6$, average degree parameter was $\nu_n = 30$, and the inter-community probability scaling matrix was $B = (\nu_n/n) \cdot \begin{pmatrix} 0.5 & 1 \\ 1 & 0.5 \end{pmatrix}$. Initial features $X_i$ were $d = 2$ dimensional drawn from mean vectors $M_{1,*} = [2,2]^T$ and $M_{2,*} = [-1,-2]^T$. Quadratic discriminant analysis was performed using the sample statistics of $\overline{\phi}_i^{(k)}$ with $k = 2$. Cross-entropy training consisted of single linear layer trained for 5000 epochs at learning rate 0.5 with a SGD optimizer

### G.4 Details for Figure 4

The plots in Figure 4 were generated in the same setting as Section G.3 with the exception of a higher average degree $\nu_n = 35$. The plots show Kernel Density Estimates (KDEs) of the $\overline{\phi}_i^{(k)}$ features for $k \in \{2,4,6\}$. The KDEs were computed using Gaussian kernels with bandwidth selected by Scott's rule.

### G.5 Details for Figure 5

The plot of Figure 5 was generated from a 2-class SBM with 32000 nodes. Class proportions were $\pi_1 = 0.9, \pi_2 = 0.1$, average degree parameter was $\nu_n = \sqrt{32000}$, and the inter-community probability scaling matrix was $B = (\nu_n/n) \cdot \begin{pmatrix} 10 & 0.1 \\ 0.1 & 10 \end{pmatrix}$. Initial features $X_i$ were $d = 1$ dimensional and generated as $X_i \sim N(M_{z_i}, \sigma^2)$ for $M_1 = 10^{-2}, M_2 = -10^{-2}$ and $\sigma^2 = 10^{-4}$.

For the plot of Figure 5 we simulate 100 CSBM graphs each at 32000 nodes. From these 100 replicates, we obtain an estimate for $\mathbb{E}[\xi^{(k)}]$ with $k = 3$. The final figure is a 100 bin histogram of the 3200000 empirical elements with a theoretical density given by our theory drawn on top.

