# OpenReview forum: "A CLT for Polynomial GNNs on Community-Based Graphs"
_NeurIPS.cc/2025/Conference — NeurIPS 2025 poster_

### Official Review · Reviewer_BzXf · 2025-06-19

**Clarity:** 3
**Significance:** 2
**Originality:** 2
**Rating:** 4
**Confidence:** 4

**Summary:**

The paper studies the asymptotic properties of the product $A^k X$ where $A$ is an adjacency matrix and $X$ is a matrix of node features under a sparse community graph model, and in doing so are able to characterize the asymptotic distribution of the embedding vectors produced through this matrix product. In particular, they are able to give guarantees for a Wasserstein-bound CLT for these features (both after being re-scaled by a suitable power of the average degree of the network, and after the typical centering/scaling one expects for CLT results). This is then used to give some discussion and insights into the behavior of using these features for classification, and the well known phenomenon of over-smoothing in GNNs.

**Questions:**

1. I am surprised that the results do not extend to the dense regime - for instance, by rewriting Assumption 3 as $( \nu_n / n)^{1/2} \cdot \sqrt{n} ( \pi_l - | \mathcal{C}_l | / n ) = o(1)$, it appears that if $\nu_n = n$, then a standard assumption of assuming that the community assignments are drawn from a Multinomial distribution would break this assumption. Can you please give some intuition as to what breaks down in this regime?

2. I am a bit confused by the presentation of Assumption 1, where the average degree parameter $\nu_n$ is said to need to grow to infinity as the number of nodes does (which is a standard assumption). There is then also the assumption that $\nu_n B_{ll'} / n \leq C/n$ for some constant $C$, which when rearranged gives that $\nu_n B_{ll'} \leq C$, which contradicts $\nu_n \to \infty$. I'm assuming this is a typo and that it just suffices to assume that $B_{ll'}$ is bounded, which under the assumptions that $L$ is fixed with $n$ is going to be the case anyways.

3. Can you please provide some intuition for why the covariance matrices in Theorem 1 and Theorem 2 have no dependence on the covariance structure of the node features $X_i$? Just thinking out loud about the variance of the node features, we know that by the law of total variance, it will break down into two terms - one corresponding to how the covariance of the node features is smoothed by the random walks on the graph of length $k$, and the other corresponding to the uncertainty in the graph mixed by the mean signal of the $X_i$. This second term is what appears in the variance bound - why does this end up being the dominating term?

4. Can you please give a summary of where your results fit into the existing literature on theoretical guarantees for GNNs? For example, there is a substantial literature already which covers the oversmoothing phenomenon in GNNs - to what extent is the discussion in this paper novel? For example, to cite some well known papers which discuss this phenomenon:

- In "Graph Neural Networks Exponentially Lose Expressive Power for Node Classification" by Oono and Suzuki, they are able to give a characterization of this phenomenon for Erdos-Renyi graphs within GCNs with ReLU activations, and are able to show subspace collapse in the learned embeddings which is achieved exponentially in the number of layers.
- In "A Note on Over-Smoothing for Graph Neural Networks" by Cai and Wang, this is extended to handle other non-linear activation functions.
- "Not too little, not too much: a theoretical analysis of graph (over)smoothing" by Keriven studies the same type of model as in the current paper, albeit with the Laplacian rather than the adjacency matrix (which is the more frequently used model).
- "Demystifying Oversmoothing in Attention-Based Graph Neural Networks" by Wu et al studies GNNs with attention mechanisms, and also handles the non-linear regime.

Can you explain the novelty in your contribution compared to these papers? To be clear, this last question is the one of main importance to me in terms of understanding the overall contribution of the work. An understanding of whether the results in this paper develop substantial **new** understanding for the behavior of GNNs will be the main factor in whether my overall evaluation would increase.

**Ethical Concerns:**

["NO or VERY MINOR ethics concerns only"]

**Final Justification:**

After the rebuttals from the authors clarifying the significance of their results, the way in which their results are distinguished from the rest of the literature (this being my main barrier to recommending the acceptance of the paper), and to also highlight the ways in which their work may be extended (using similar approaches) to models of closer interest to practitioners, I am happy to recommend the acceptance of the paper.

**Limitations:**

The authors have adequately addressed the societal impact of their work. As it comes to the limitations of the theoretical analysis, there is not really much of a discussion at all except a sentence at the very end of the conclusion referring to non-linear GNNs and different graph models. I would appreciate a more involved discussion about a) why their results cannot be extended easily to non-linear GNNs, GNNs with learnable weight matrices within layers, GNNs with Laplacian matrices, and so on; and b) a discussion about what aspects of the theory break down when moved to the either the dense regime where $\nu_n = \Omega(n)$ or the regime where $\nu_n = O(1)$. This would improve the overall contribution of the paper.

**Paper Formatting Concerns:**

I have no paper formatting concerns.

**Quality:**

3

**Strengths And Weaknesses:**

Strengths of the paper:
The results in the paper are quite technical in their nature (for instance, requiring some detailed analysis into random walks of graphs) and are generally presented well and are of high quality. The mathematical details in the appendix are correct, and are covered at an appropriate level of detail. From a probabilistic perspective, the results of the paper are interesting. To my knowledge, the mathematical results in of themselves are new.

Weaknesses of the paper:
It is generally unclear how this paper fits into the larger literature on GNNs and the phenomenon which the paper seeks to explain, and the extent to which it is novel is unclear. The GNN model being studied is also relatively simple - even ignoring the removal of the non-linear activation function between layers, the features being studied are simply node features propagated by a power of the adjacency matrix. With the exception of Section 4.1 (where there is only a discussion of a linear model applied to the features), there is no reference to trainable features for this model. Moreover, there are already a large number of papers which study the GNN oversmoothing phenomenon, some of which already give explanations under more realistic assumptions than the current paper. The overall novel contribution in terms of understanding these models from a ML perspective appears to be minimal.

---

> ### Author Rebuttal · Authors · 2025-07-31
>
> > 1. I am surprised that the results do not extend to the dense regime - for instance, by rewriting Assumption 3 as , it appears that if , then a standard assumption of assuming that the community assignments are drawn from a Multinomial distribution would break this assumption. Can you please give some intuition as to what breaks down in this regime?
>
> Assumption 3 can be relaxed, as we have already done in the supplementary material (we will clarify this in the revision). The assumption can be weakened to $|C_\ell| / n - \pi_\ell = o(1)$, which holds even when $\nu_n=n$ for balanced community assignments.
>
> However, other parts of our argument rely on the sparse regime, $\nu_n = o(n)$. Specifically, in this regime, the graph noise is the dominant source of variance, and the initial feature noise can be ignored in the limit. This is not the case in the dense regime ($\nu_n = \Omega(n)$), where the feature noise is no longer negligible. In the dense case, the limiting covariance matrix would need to be modified to account for both sources of noise. We elaborate on this in the rebuttal for reviewer SM6e.
>
> > 2. I am a bit confused by the presentation of Assumption 1, where the average degree parameter is said to need to grow to infinity as the number of nodes does (which is a standard assumption). There is then also the assumption that for some constant , which when rearranged gives that , which contradicts . I'm assuming this is a typo and that it just suffices to assume that is bounded, which under the assumptions that is fixed with is going to be the case anyways.
>
> Thank you for pointing this out, and our apologies for the confusion. This is indeed a typo. The condition should be $B_{\ell\ell'} \le C$, meaning the entries of the affinity matrix $B$ are bounded. We will correct this in the revision.
>
> > 3. Can you please provide some intuition for why the covariance matrices in Theorem 1 and Theorem 2 have no dependence on the covariance structure of the node features ? Just thinking out loud about the variance of the node features, we know that by the law of total variance, it will break down into two terms - one corresponding to how the covariance of the node features is smoothed by the random walks on the graph of length , and the other corresponding to the uncertainty in the graph mixed by the mean signal of the . This second term is what appears in the variance bound - why does this end up being the dominating term?
>
> This is an excellent question, and your intuition is correct. The total variance indeed breaks down into two terms, which correspond to the graph noise ($\Delta^\circ$ in our proof) and the feature noise ($\Lambda^\circ$). See Equation (22) in the Supplement.
>
> Our analysis shows that graph noise dominates in the sparse regime we consider (i.e., as $\nu_n \to \infty$ with $\nu_n = o(n)$). In the dense regime ($\nu_n = \Omega(n)$), however, the feature noise is no longer negligible and contributes to the limiting variance.
>
> Why this happens? We don't have a simple answer;  it is a direct consequence of our careful moment analysis. By characterizing the dominant walk structures, we precisely determine the asymptotic order of these two noise terms. Please see Propositions 2 and 3 and Lemma 5 of the supplementary material. We also touch upon this in rebuttal to reviewer SM6e.
>
> > 1. Can you please give a summary of where your results fit into the existing literature on theoretical guarantees for GNNs?
>
> There is a **straighforward explanation** for oversmoothing via **power iteration**: If you take a fixed, known positive matrix $A$ and iterate it to get $A^k$. Then, as $k$ gets large ($k \to \infty$), the operator collapses towards a rank-one operator in the direction of the Perron eigenvector. In other words, power iteration uncovers the leading eigenvector of matrix. This is not very surprising. The work you cite (including the nonlinear extensions) provide this type of explanation.
>
> Our work differs fundamentally by considering a **stochastic setting**. Here, the aggregation operator $A$ is a noisy, random matrix, not a fixed one. We analyze the limit as the graph size $n \to \infty$ for a *fixed* depth $k$. In this asymptotic regime, a Central Limit Theorem emerges.
>
> While the power iteration analogy still applies to the first-order term (the mean signal), our CLT also characterizes the second-order effects. Crucially, we show that the covariance *also* collapses (via a term related to power iteration) in the same direction as the signal. This means that for sparse graphs, the potentially high-dimensional nature of the features is lost in the limit—a much stronger statement than standard oversmoothing results. Characterizing this behavior as $n \to \infty$ under a stochastic model is a key distinction from prior work. Please see the rebuttal to reviewer SM6e for more context on our work relative to the current literature.
>
> **Limitation**
>
> We will add a thorough discussion of limitations to the revised manuscript. In short:
> *   **Dense Regime:** There is no fundamental obstacle to applying our techniques to the dense regime ($\nu_n = \Omega(n)$), though the analysis would need to be extended to account for both graph and feature noise sources, as mentioned earlier.
> *   **Nonlinearities:** A significant challenge is extending our walk-based analysis to GNNs with nonlinear activation functions. In response to Reviewer ovye, we outline some potential ideas. However, developing entirely new methods for deriving CLTs for GNNs might be a more fruitful long-term direction. We see our work as a first step, establishing a foundation for more general and easily applicable techniques.
> *   **Attention/Trainable Weights:** Extending the analysis to models like GAT with trainable aggregation weights is another direction. While potentially feasible, it would require a substantial and non-trivial extension of the walk analysis.

---

> ### Comment · Reviewer_BzXf · 2025-07-31
>
> Thank you for the detailed response to my questions and concerns. Re: question 3, it would be nice if there were a nice intuitive explanation, but sometimes the math avoids this - so it goes.
>
> With regards to my Question 4 and having read the rebuttal to SM6e (with regards to how the results of the paper fit into the wider literature), I think the key contribution to highlight (perhaps even moreso than the stochasticity) is the following:
>
> > Crucially, we show that the covariance also collapses (via a term related to power iteration) in the same direction as the signal. This means that for sparse graphs, the potentially high-dimensional nature of the features is lost in the limit—a much stronger statement than standard oversmoothing results.
>
> With the rebuttal response to SM6e and myself in mind, I am satisfied that there is some novelty with regards to the insights produced by the results in the paper, and will increase my scoring to one in the accept range as a result. I hope that the final version provides some discussion about the novel contributions of the paper with respect to the wider literature, in addition to the some of the discussions provided to the other reviewers with regards to ways in which the results could be extended (if albeit tedious) vs ones requiring additional machinery. I believe the results of the paper would be of broader significance to practitioners as a result of this.

---

> > ### Author Response · Authors · 2025-08-08
> >
> > Thank you for the thoughtful follow-up and for raising your score. We’ll make the covariance-collapse mechanism a central takeaway as you suggest, and clearly distinguish it from standard oversmoothing. In the camera-ready we’ll expand the related-work discussion to situate this contribution, and briefly note which extensions are straightforward (though tedious) versus those that need new machinery.

---

### Official Review · Reviewer_SM6e · 2025-06-30

**Clarity:** 4
**Significance:** 3
**Originality:** 2
**Rating:** 5
**Confidence:** 4

**Summary:**

In this paper, the authors show two CLT-type convergence results for linear GNNs on CSBM random graphs when the number of nodes go to infinity, and the graph is sparse. The first result pertains to rescaled node features, which they show converge to a mixture of Gaussians with vanishing covariances. The second result concerns a centered and again-rescaled version of these features, which they show converge to a sum of centered Gaussians with specific covariances. They show two applications, one concerning the training of classifiers on these features for which they give a uniform deviation result, and one concerning some remarks on oversmoothing, when the number of layers grows. The proofs rely mostly on moment computations.

**Questions:**

See above for several questions.

**Ethical Concerns:**

["NO or VERY MINOR ethics concerns only"]

**Final Justification:**

The detailed answers by the authors strengthened my positive view of their results. I have increased my score, understanding that the interesting discussion in their answers will be added to some extent to the final version of the paper.

**Limitations:**

Limitations are appropriate.

**Quality:**

3

**Strengths And Weaknesses:**

Strength
- the paper is well-written and clear, despite heavily technical content
- one notable strength of the results is that, as CLTs, the limit does not depend on the distribution of the input node features (only on its mean), which is quite original in this literature. Is this due to the sparsity? (looking at the proof it allows to discard some terms)
- beyond the convergence results themselves, the applications on training and oversmoothing are interesting

Weaknesses
- the references are quite scarce, given this large literature, especially on the oversmoothing aspect. Several papers performed similar theoretical analyses before (with shrinking Gaussian communities, collapsing covariances, and so on), and while that of the authors is new (as far as I know), it must commented with regard to these previous results. It would help better outline the novelty and strength of the results: while the current formulation gives an impression of novel "idea" that is relatively incorrect, it is rather a new take on a classical idea. See eg [1,2,3] (non-exhaustive list).

In the same vein, linear GNNs are very often called "Simplified Graph Convolution" (SGC), as per the seminal paper [4]. To my knowledge, polynomial GNNs are more often used for polynomial filters, where every power of the graph matrix (eg Chebnet).

[1] Baranwal et al. "Effects of Graph Convolutions in Multi-layer Networks"
[2] Keriven "Not too little, not too much: a theoretical analysis of graph (over)smoothing"
[3] Wu et al "A Non-Asymptotic Analysis of Oversmoothing in Graph Neural Networks"
[4] Wu et al. "Simplifying Graph Convolutional Networks"

- could the result on training linear GNNs be pushed further? It feels a bit "unfinished" at the moment. Surely the limit problem is a known classification problem, with guarantees?
- while Theorem 2 indeed feels like a CLT, this is not quite the case of Theorem 1, which uses vanishing covariances. Unless I am missing something, a Gaussian with vanishing covariance converges to a Dirac in Wasserstein norm, does that means that the distribution really converges to a sum of Diracs? What would be the implication of this for the other results? (training, oversmoothing). This may be discussed
- In theorem 2 the centering of the features uses their unknown mean. Could it be replaced by some kind of empirical average? Otherwise it must be clearly written that this is a theoretical result that uses oracle quantities.
- the authors mention the importance of sparsity, but that "some" results may carry to the dense setting. Is it possible to provide a bit more details what would not carry, and why?
- minor comment: it is just a personal appreciation, but I feel like some of the language used by the authors oversells a bit the results; eg I'm not sure a result pertaining to linear GNNs on sparse CSBM can be said to have "profound implications for the training of GNNs" in general. Paradoxically, this is a bit detrimental to the just appreciation of the results, which I find to be valuable and carry interesting intuitions by themselves.

---

> ### Author Rebuttal · Authors · 2025-07-31
>
> > The references are quite scarce, given this large literature, especially on the oversmoothing aspect. Several papers performed similar theoretical analyses before (with shrinking Gaussian communities, collapsing covariances, and so on), and while that of the authors is new (as far as I know), it must commented with regard to these previous results.
>
> In our work, we establish a limiting distributional result for multi-hop aggregated node features on community-structured graphs. Our result, along with its closed-form characterization, is novel. Moreover, its mode of convergence is unique, the convergence in expected Wasserstein distance enables high-probability guarantees for Lipschitz optimization problems involving multi-hop aggregated node features. We emphasize that this is much stronger than the usual "convergence in distribution" result common with CLTs.
>
> The related literature for multi-hop node features can be broken into three categories: distributional characterizations, oversmoothing phenomenon, and improvement results for select learning tasks (commonly classification or regression).
>
> For **distributional characterizations**, [1] is closest to the work of the authors. In [1], the results hinge on the statement that $\phi^k_i$ is exactly component-wise Gaussian. Note, that this cannot be the case as even $\phi_i^{(1)} = \sum_{j}A_{ij}x_j$ with Bernoulli $A_{ij}$ and normal $x_j$ is, by definition, a (scaled) mixture of Gaussians.
>
> In the vein of **oversmoothing**, works [2], [3], and [4] show how properly normalized aggregations can still oversmooth in the presence of non-linearities. Oversmoothing can be seen as a consequence of the power iteration $A^k$ collapsing the range onto the Perron eigenvectors. The works [2] and [3] show that non-linearities like ReLU do not help oversmoothing since the ReLU operator is also contractive under the operator norm (see [5]). In [4] the authors extend these results to include to also include attention-based non-linearities. Outside of [2], which considers the effects of oversmoothing on a $L=1$ community graph, the results of all other works assume a deterministic graph.
>
> With respect to **improving task performance**, works [6] and [7] show how multi-hop features can improve downstream learning tasks. Between the two works the generative formulation differs, [6] assumes $(p,q)$-SBM with mixed mean feature representations while [7] assumes a low rank, latent variable model which yields dense observed graphs. The losses considered by [6] and [7] are Lipschitz, highlighting the importance of understanding behavior of multi-hop features under the 1-Wasserstein metric.
>
> > In the same vein, linear GNNs are very often called "Simplified Graph Convolution" (SGC), as per the seminal paper [4]. To my knowledge, polynomial GNNs are more often used for polynomial filters, where every power of the graph matrix (eg Chebnet).
>
> Yes. Our original thought was that, since we provide the limit for every $k$, it should be straightforward to derive similar results for polynomial GNNs of the form $\sum_{k=1}^{K} A^{k} X W_{k}$, which includes ChebNet and the like. Upon further reflection, we realized that there are still complications to resolve—specifically, the statistical dependence between $A^{k} X$ and $A^{k'} X$ for $k \neq k'$. Nevertheless, we believe our approach can be extended (with some tedious bookkeeping) to derive CLTs for these models.
>
> That said, we can rename the present paper “monomial GNNs” if you feel strongly about it. “SGC” is not very descriptive, and we believe “polynomial” or “monomial” is more appropriate.
>
> > could the result on training linear GNNs be pushed further? It feels a bit "unfinished" at the moment. Surely the limit problem is a known classification problem, with guarantees?
>
> The discussion of training linear GNNs with cross-entropy is meant only as an illustrative example. Indeed, we comment (Lines 266-268) that it is generally **sub-optimal**: unless the asymptotic covariances are equal, the Bayes decision boundaries are quadratic, so any linear classifier will incur a (typically small) excess risk. Highlighting this gap is one insight of our analysis.
>
> Nothing in our framework restricts us to linear decision rules or to the CE loss. One could instead fit $f(\bar\phi_i^{(k)})$ using richer function classes—quadratics, higher-order polynomials, RKHS functions, etc.—and our theory still applies so long as the composed "loss $\circ f$"  is Lipschitz. Each choice maps to a corresponding (simpler) population-level optimisation problem in the limit.
>
> Finally, when the loss is convex, classical results establish conditions under which minimising that convex surrogate matches the Bayes-optimal zero-one risk. See, for example,
>
> Bartlett, P. L., Jordan, M. I., & McAuliffe, J. D. (2006). Convexity, classification, and risk bounds. Journal of the American Statistical Association, 101(473), 138–156.
>
> We will cite this work in the revision; a deeper treatment lies beyond the scope of the present paper.
>
> > Unless I am missing something, a Gaussian with vanishing covariance converges to a Dirac in Wasserstein norm, does that means that the distribution really converges to a sum of Diracs?
>
> Thank you for the observation. You are correct: as the covariance matrix vanishes, a Gaussian collapses to a Dirac mass. However, for any finite (even moderately large) n, the covariance is still positive (as it is scaled by $\nu_n$ which could grow very slowly), so the Gaussian approximation retains valuable second-order information that the Dirac limit loses. We therefore interpret Theorem 1 as a finite-n approximation; the Dirac point mass is merely the asymptotic endpoint, while the Gaussian mixture captures the covariances that matter in practice.
>
> Figure 4 in Appendix G illustrates this behavior: with an average degree as low as $\nu_n = 16$, the empirical distributions already match the Gaussian prediction, and the covariances remain far from negligible.
>
> > In theorem 2 the centering of the features uses their unknown mean. Could it be replaced by some kind of empirical average?
>
> The CLT exists irrespective of whether we can estimate $\mathbb{E}[\phi]$ in a finite sample setting. Estimating $\mathbb{E}[\phi]$ is closely related to correct identification of the labels $z$, which is the goal in the first place.
>
> >  Otherwise it must be clearly written that this is a theoretical result that uses oracle quantities.
>
> We would not describe our result as relying on “oracle” quantities. It is simply a central-limit-type statement.  Consider the classical CLT: We have i.i.d. variables $X_1,\dots,X_n$ with unknown mean $\mu$ and the CLT says
> $\sqrt{n} (\bar X_n - \mu) \to N(0,1).$
> Although $\mu$ is unknown, no one regards the theorem as “oracle-based”. The CLT in fact is used to give uncertainty bounds on the unknown $\mu$ based on the empirical mean $\bar X_n$. Now, replace $\bar X_n$ with $\bar \phi_i^{(k)}$ and $\mu$ with $\mathbb E \bar \phi_i^{(k)}$ and you get a similar interpretation in our case (with a bit of nuance since we look at the empirical distribution of these after scaling, a step which is not present in classical CLT.)
>
> > The authors mention the importance of sparsity, but that "some" results may carry to the dense setting. Is it possible to provide a bit more details what would not carry, and why?
>
> The limiting CLT still holds for the dense regime although the limiting characterization of the GNN may differ. Section A.1 gives a detailed breakdown of graph and feature noise ($\Delta$ and $\Lambda$ resp.) and how these terms contribute to the overall noise characteristic of the centered aggregated features $\phi^{(k)} - \mathbb{E}[\phi^{(k)}]$. The scale of both the graph noise and the feature noise is given in Section A.3. Here, we see that the feature noise scale depends on a $\beta_{k,n}$ parameter. As shown in Lemma 5 of the supplement, this parameter goes to zero whenever $\nu_n/n\to 0$ and $\nu_n \to\infty$.
>
> In the dense regime, the feature noise $\Lambda$ makes non-negligible contributions to the empirical distribution's sliced moments $\langle \xi_i^{(k)},\theta\rangle^r$. As a result, one would need to account for $\Lambda_{i,\theta}^r$ in approximation Lemma 5. This, in turn, would have downstream impacts on the closed form of the limiting covariance (Proposition 4 of the supplement).
>
> That being said, while the addition of non-negligible feature noise may complicate the final characterizations of the limiting covariance, there does not seem to be any hard technical barriers to showing an analogous CLT result for the dense graph case.
>
> > I feel like some of the language used by the authors oversells a bit the results
>
> This language will be amended appropriately in the final revision of our paper.
>
>
> [1]  Wu, X., Chen, Z., Wang, W., & Jadbabaie, A. (2022). A non-asymptotic analysis of oversmoothing in graph neural networks.
>
> [2] Oono, K., & Suzuki, T. (2020). Graph neural networks exponentially lose expressive power for node classification.
>
> [3] Cai, C., & Wang, Y. (2020). A note on over‑smoothing for graph neural networks.
>
> [4] Wu, X., Ajorlou, A., Wu, Z., & Jadbabaie, A. (2023). Demystifying oversmoothing in attention‑based graph neural networks.
>
> [5] Dittmer, S., King, E. J., & Maass, P. (2018). Singular values for ReLU layers.
>
> [6] Baranwal, A., Fountoulakis, K., & Jagannath, A. (2022). Effects of graph convolutions in multi‑layer networks.
>
> [7] Keriven, N. (2022). Not too little, not too much: a theoretical analysis of graph (over)smoothing.
>
> [8] Baranwal, A., Fountoulakis, K., & Jagannath, A.. (2021). Graph Convolution for Semi-Supervised Classification: Improved Linear Separability and Out-of-Distribution Generalization.

---

> > ### Comment · Reviewer_SM6e · 2025-08-04
> >
> > I thank the authors for their detailed answers, which strengthened my positive view of their results. Changing the title of the paper is absolutely not a requirement from my part. I have increased my score, understanding that the interesting discussions in their answers will be added to some extent to the final version of the paper.

---

> > > ### Author Response · Authors · 2025-08-08
> > >
> > > Thank you for the thoughtful follow-up and for increasing your score. We appreciate the clarification that a title change isn’t required, and we will incorporate the key discussions from our responses into the final version.

---

### Official Review · Reviewer_vVTK · 2025-07-01

**Clarity:** 4
**Significance:** 2
**Originality:** 2
**Rating:** 4
**Confidence:** 4

**Summary:**

This paper establishes CLTs for the features of a particular class of GNNs when the input data follow a form of stochastic block model. The limits distributions in the CLTs are mixtures of Gaussians with parameters expressed in terms of the ingredients of the block model. Insights into the performance of linear classification and the over smoothing phenomenon are derived from the CLTs and the theory is illustrated with numerical experiments.

**Questions:**

1) The work in this paper has the potential to stimulate similar analysis of GNN's other than Poly-GNN's, can the authors very briefly comment on what the main opportunities and obstacles might be? If they need to save some space to accommodate this, I would recommend condensing section 1.1, I think it could easily be half the length and still do the job of summarising the existing contributions.

2) Regarding the place of this work within the literature, I think more should be said about the current state of theory for GNN's and poly GNN's in particular. Are there any existing CLT's, or other theory related to behaviour of GNN's under block models etc? If, in any sense, the results in this paper are the first of their kind, I think that must be highlighted. More broadly to frame originality, what was the inspiration for this work? Is it based on any existing analysis, or something in a completely new direction? I think these questions should be answered in the paper, not just in the response to the review.

3) A technical question, related to reqs (10), (11) and elsewhere: inside the expectation their is a sup over a (presumably) uncountable family of random variables (associated with each of the Lipschitz functions). Consider this uncountable sup, how do you know the sup is measurable, and hence well-defined as a random variable?

4) Line 230 "...significant practical implications..." On first reading this, I was expecting that the theory would be use derive some guidance or new techniques for how GNNs should be use in practice. I'm not sure I got that, but rather some (detailed) insights into phenomena that might be observed in practice. The authors may consider re-wording slightly here to manage the readers' expectations.

5) Line 238 "...as these are the quantities typically used by the classifier” Please clarify - the degree normalised features depend on nu_n; even if one accepts the assumption that the data are generated by the block model, it seems too much of a stretch to assume that the true value of nu_n is known.

6) Line 272 "...is a well-documented empirical observation” Please add references to the literature.

A couple of typos:

Abstract "Our results provides a..."

Figure 1 caption “Ten gradients steps…”

**Ethical Concerns:**

["NO or VERY MINOR ethics concerns only"]

**Final Justification:**

I have upped my score to 4 to reflect the fact I find the authors’ response satisfactory.

**Limitations:**

The authors have answered “No” to the question of whether limitations are discussed. This feels like a missed opportunity!! Surely there must be something you can say here?!

They have also answered “No” to the question about open access to data and code. Whilst I agree the numerical experiments are not a focal point of the paper, it seems an unnecessary restriction to not make it easy for the community to reproduce or further experiment with the numerical examples. They have also answered “No” to the question about compute resources. Again this feels unnecessary: just add it as a brief comment in the supplementary material.

The authors have answered “No” to the question about societal impacts. This is not inappropriate given the theoretical nature of the contribution.

**Quality:**

4

**Strengths And Weaknesses:**

From a rigorous mathematical point of view, GNNs have remained a somewhat murky subject. In the ML literature, even the motivation for GNN’s is often not well explained beyond simple heuristics, and their has been an absence of careful statistical formulation of GNN’s operation, nevermind statistical theory of the sort set out in this paper. As such I believe the work set out in this paper could make an important step forward and stimulate more work in this area.

The mathematical clarity of the paper is very good. I have experience in statistical theory of networks and graphs, but am not an expert in GNNs; for me the presentation of theory made it easy to digest.

The paper focuses on one particular class of GNN’s: poly-GNN’s, which have a relatively simple linear structure.
My questions below include a request to clarify exactly where this paper sits in the literature on GNN’s, its originality,  and for comments to be added on the prospects of the theory being extended beyond GNN’s. Subject to that being clarified, I think it would be wrong to regard the focus on poly-GNN’s as a “weakness” of the paper, so much as a natural limitation in its scope.

I think there are some opportunities for other small improvements — see questions below.

If the authors can further clarify the originality of this work and its position in the literature about the theory of GNNs I am willing to up my scores about significance and originality, and hence boost my overall score for the paper. Clarification of originality and place in the literature is of course a very important matter, so without this addressed I have currently rated as "borderline reject".

---

> ### Author Rebuttal · Authors · 2025-07-31
>
> > 1. The work in this paper has the potential to stimulate similar analysis of GNN's other than Poly-GNN's, can the authors very briefly comment on what the main opportunities and obstacles might be? If they need to save some space to accommodate this, I would recommend condensing section 1.1, I think it could easily be half the length and still do the job of summarising the existing contributions.
>
> Thank you for the positive comment. We agree that our work can stimulate similar analyses of other GNNs.
>
> **Main Opportunities:**
> 1.  **Feasibility and Guidance:** By demonstrating that a CLT exists for Poly-GNNs, we establish its feasibility for the broader GNN family. Once you show that the CLT exists for one family, there is a high chance that it exists for others. Our theoretical framework, particularly the characterization of the CLT's scale (normalization by $\nu_n^{k-1/2}$) and the quantity to look at (empirical distribution of the GNN embeddings), provides a roadmap for researchers seeking to develop new CLTs for other architectures.
> 2.  **Stochastic Analysis:** We show that studying GNNs under a stochastic network model (asymptotically as $n \to \infty$ in the sparse regime) is a fruitful approach. This allows us to precisely characterize the effect of graph noise and observe non-trivial phenomena: the original feature covariance vanishes and is replaced by a purely graph-induced covariance, which in turn collapses in the "worst possible" direction (the same as the signal). We hope this sparks more interest in the stochastic analysis of GNNs, moving beyond the common assumption of a known, noiseless graph.
> 3.  **Optimization in the Limit:** Our use of Wasserstein-type bounds, which implies asymptotic convergence for all Lipschitz functions, shows that the entire cross-entropy optimization objective can be studied in the limit. This may inspire others to seek Wasserstein-type CLTs when studying the optimization aspects of GNNs.
> 4.  **A Generalizable Technique:** The walk-analysis technique, while complex, reduces the problem to careful combinatorics and allows for sharp control of the moments. This technique could potentially be used to derive a CLT for GNNs of the form $\sum_{k=1}^K A^k X W_k$, revealing interesting interactions between aggregations at different layers.
>
> **Main Obstacle:**
> The primary obstacle is the walk analysis itself. It is best suited for polynomial aggregation, though an extension to gated networks might be possible (see our response to Reviewer ovye). However, the method of moments we employ is just one approach to deriving CLTs. We believe other, more general approaches may exist that, once discovered, could allow for the derivation of CLTs for many GNN families from a few foundational results.
>
> > 1. Regarding the place of this work within the literature, I think more should be said about the current state of theory for GNN's and poly GNN's in particular. Are there any existing CLT's, or other theory related to behaviour of GNN's under block models etc? If, in any sense, the results in this paper are the first of their kind, I think that must be highlighted. More broadly to frame originality, what was the inspiration for this work? Is it based on any existing analysis, or something in a completely new direction? I think these questions should be answered in the paper, not just in the response to the review.
>
> We discuss the literature further in response to Reviewers SM6e and BzXf and will expand on this in the revision.
>
> To briefly answer your questions: to our knowledge, there are no other CLTs for GNNs in the literature. While theoretical analyses of GNNs exist, most are non-stochastic, assuming a fixed, given (and often "true") graph. Treating the connections as stochastic (i.e., noisy) is more realistic and adds significant nuance. The few works that consider stochastic models (like SBMs) often provide non-rigorous analysis (e.g., physics-type arguments) or are limited in scope (e.g., to $k=1$ or bounds on misclassification error). They do not provide an exact characterization of the asymptotic distribution as we do here.
>
> Yes, we believe our results are the first of their kind and will highlight this in the revision.
>
> The inspiration for this paper comes from a previous work of ours, which we cannot cite due to anonymity. In that work, we sought to understand the effect of depth in GNNs (i.e., is there an optimal depth $k$?). We studied the signal-to-noise ratio (SNR) of the GNN output and demonstrated that the rate at which the SNR increases with $n$ is independent of depth (the signal grows like $\nu_n^k$ while the noise grows like $\nu_n^{k-1/2}$, so $k$ cancels in the ratio). The walk analysis we used there was sharp enough (providing tight upper and lower bounds) that we realized we could exactly characterize the limiting moments. This observation led to the current paper, though many new details needed to be worked out (e.g., controlling moment variances, characterizing asymptotic covariance, and concentrating the empirical distribution). We believe the two papers together chart a new direction and will add a discussion along these lines to the revision once the anonymity barrier is lifted.
>
> > 2. A technical question, related to reqs (10), (11) and elsewhere: inside the expectation their is a sup over a (presumably) uncountable family of random variables (associated with each of the Lipschitz functions). Consider this uncountable sup, how do you know the sup is measurable, and hence well-defined as a random variable?
>
> Thank you for this interesting technical question. The supremum is indeed well-defined because it can be reduced to a supremum over a countable subset, ensuring measurability.
>
> The argument is as follows: for any 1-Lipschitz function $f$, one can construct a sequence of functions $\\{\phi_m\\}\_{m\ge 1}$ where each $\phi_m$ matches $f$ on $[-m, m]$ and extends linearly with slope $\pm 1$ outside this interval. The difference of the two measures on $f$ will be the same as $\sup_{m \ge 1}$ of the difference over $\phi_m$.  Furthermore, for any given $m$, the space of 1-Lipschitz functions on the compact interval $[-m, m]$ (equipped with the sup-norm) is totally bounded and thus separable. It admits a countable dense subset, say $\mathcal{F}\_m$. Therefore, the original supremum over an uncountable set is equal to the supremum over the countable set $\bigcup_{m \ge 1} \mathcal{F}_m$.
>
> Similar approximation arguments are used in the proof of Proposition 8 in Appendix E.
>
> For completeness, we note there are other general approaches to handle such measurability issues: (1) redefining the supremum, as in Talagrand's *The Generic Chaining* (Equation (0.2)), or (2) using outer expectation in place of $\mathbb{E}$, as in the work of van der Vaart and Wellner.
>
> > 3. Line 230 "...significant practical implications..." On first reading this, I was expecting that the theory would be use derive some guidance or new techniques for how GNNs should be use in practice. I'm not sure I got that, but rather some (detailed) insights into phenomena that might be observed in practice. The authors may consider re-wording slightly here to manage the readers' expectations.
>
> Will do. Thank you for the suggestion.
>
> > 4. Line 238 "...as these are the quantities typically used by the classifier” Please clarify - the degree normalised features depend on nu_n; even if one accepts the assumption that the data are generated by the block model, it seems too much of a stretch to assume that the true value of nu_n is known.
>
> That is a fair point. But, the average degree of the observed graph provides is a very good empirical approximation of $\nu_n$ , up to a constant. Since only the scaling matters for the normalization, using the observed average degree is a practical and effective approach.
>
> > 5. Line 272 "...is a well-documented empirical observation” Please add references to the literature.
>
> This will be done. We have also added a brief review of the oversmoothing literature in responses to Reviewers SM6e and BzXf.
>
> Regarding the limitations, we will add a discussion of them in the revision.  Key points:
>
> - **Scope of the theory.** Our results characterize multi-hop features optimized under Lipschitz losses. They exclude nonlinear propagation and other complex GNN variants; extending to those architectures is future work.
> - **Learnable components.** Trainable functions can act on the aggregated features (the “head”), but the propagation operator itself is fixed. We will clarify this constraint.
> - **Dependence on average degree $\nu_n$.** The theory holds for $\nu_n \to \infty$ up to to $\nu_n = o(n)$; the unrealistic $\nu_n = O(1)$ regime is excluded, and $\nu_n = \Omega(n)$ requires additional arguments. Empirically, asymptotic behavior appears once $\nu_n \gtrsim 16$ (Appendix G).
> - **Model class.** We assume an SBM. While this omits truly arbitrary graphs, SBMs with large $K$ are universal $L^{2}$-graphon approximators [4–6], so the restriction is less severe than it first seems.
> - **Asymptotic vs. finite $n$.** Theorems are asymptotic; finite-sample deviations are possible. Nonetheless, experiments show excellent agreement even at modest $n$, with accuracy driven more by $\nu_n$ than by graph size.
> ---
>
> [4] Wolfe, P. J., & Olhede, S. C. (2013). Non-parametric graphon estimation.
>
> [5] Orbanz, P., & Roy, D. M. (2015). Bayesian Models of Graphs, Arrays and Other Exchangeable Random Structures. IEEE Transactions on Pattern Analysis and Machine Intelligence, 37(2), 437–461.
>
> [6] Gao, C., Lu, Y., & Zhou, H. H. (2015). Rate-optimal graphon estimation. Annals of Statistics, 43(2), 762–785.
>
> Thanks for catching the typos. They will be fixed.

---

> > ### Comment · Reviewer_vVTK · 2025-08-01
> >
> > Thanks for the rebuttal, all my points have been addressed.

---

> > > ### Author Response · Authors · 2025-08-08
> > >
> > > Thank you for the follow-up; we’re glad the rebuttal addressed your points. We’ll incorporate the relevant clarifications into the final version.

---

### Official Review · Reviewer_ovye · 2025-07-05

**Clarity:** 3
**Significance:** 3
**Originality:** 3
**Rating:** 4
**Confidence:** 4

**Summary:**

This paper provides a rigorous theoretical analysis of the asymptotic behavior of k-layer Polynomial Graph Neural Networks (Poly-GNNs) on sparse community-based graphs, such as those modeled by the Contextual Stochastic Block Model (CSBM). The authors derive two central limit theorems (CLTs) characterizing the empirical distributions of Poly-GNN embeddings:
Theorem 1 proves that the joint distribution of degree-normalized node embeddings and class labels converges to a mixture of Gaussians with vanishing covariances.
Theorem 2 shows that the centered and scaled embeddings converge to a stable mixture of multivariate Gaussian distributions.
The paper also analytically characterizes mean and covariance collapse under repeated aggregation, offering a precise mathematical explanation for the oversmoothing phenomenon observed in deep GNNs. A key contribution is the demonstration that training a linear classifier with cross-entropy loss on Poly-GNN features is asymptotically equivalent to training on the limiting Gaussian mixture, offering theoretical insight into GNN training dynamics.

**Questions:**

Applicability Beyond Poly-GNNs: Can the authors comment on how the analysis might extend to nonlinear or weighted GNNs (e.g., GCN, GAT, GPR-GNN)? What technical barriers exist?
Sensitivity to νₙ Assumptions: Many results hinge on νₙ → ∞ and νₙ = o(n). Are the convergence rates or limiting distributions significantly affected if νₙ remains constant or grows slowly?
Practical Implications for Model Design: Given that the class means collapse to a 1D subspace, can these results motivate architectural changes (e.g., early stopping, skip connections, regularization)?
Empirical Validation: Would the authors consider adding numerical simulations (beyond Figures 1–3) to illustrate how closely the empirical distributions of embeddings follow the theoretical Gaussian mixtures, especially for moderate n?
Beyond CSBM: The authors mention the challenge of generalizing beyond community-based graphs. Can they speculate on potential extensions or possible failures of the current CLTs in graphs with heavy-tailed degree distributions or assortative mixing?

**Ethical Concerns:**

["NO or VERY MINOR ethics concerns only"]

**Limitations:**

No — The paper does not adequately discuss limitations, although the authors indirectly allude to model constraints. I recommend:
Explicitly stating the limitations of the Poly-GNN assumption (e.g., no learnable weights or non-linearities).
Discussing how sensitivity to νₙ or non-community-based graphs may affect the validity of the results.
Considering the gap between asymptotic and finite-n behavior, especially since real graphs are not infinitely large or cleanly community-structured.

**Paper Formatting Concerns:**

No formatting issues in this paper.

**Quality:**

3

**Strengths And Weaknesses:**

Quality
Strengths:
The mathematical exposition is rigorous and detailed.
The CLTs are proven with clear assumptions, and proof sketches are well-motivated.
Explicit formulas for limiting means and covariances are derived, which are novel and impactful.
Weaknesses:
No experimental validation beyond illustrative figures; this limits the practical understanding of how well the theoretical predictions manifest in real datasets.
Assumptions (e.g., community-based graph structure and νₙ → ∞) may be restrictive in real-world applications.

Clarity
Strengths:
The paper is very well written and logically structured.
Each section flows naturally into the next, and technical results are motivated with clear intuition.
The discussion on oversmoothing via spectral analysis is particularly enlightening.
Weaknesses:
Some parts of the proof outline (e.g., sub-Gaussian arguments, moment bounds) assume familiarity with advanced probability without intuitive explanation.
Notation can be heavy for non-theory audiences, especially in the oversmoothing section.

Significance
Strengths:
Offers an original, quantitative understanding of GNN oversmoothing—previously treated heuristically in much literature.
Bridges the gap between feature propagation theory and classifier training behavior.
Weaknesses:
Applicability is currently limited to polynomial GNNs on stochastic block models, which may reduce perceived generality.
Impact on GNN design may be indirect unless extended to more expressive models.

Originality
Strengths:
The use of CLTs to characterize GNN embeddings under label-aware graph models is novel.
Provides, to my knowledge, the first exact analytical treatment of Poly-GNN oversmoothing via spectral power iteration arguments.
Weaknesses:
The Poly-GNN class considered is a simplification of practical GNNs (e.g., lacks nonlinearity or learnable weights), which might limit originality for practitioners.

---

> ### Author Rebuttal · Authors · 2025-07-31
>
> >  Weaknesses: The Poly-GNN class considered is a simplification of practical GNNs (e.g., lacks nonlinearity or learnable weights), which might limit originality for practitioners.
> > Can the authors comment on how the analysis might extend to nonlinear or weighted GNNs (e.g., GCN, GAT, GPR-GNN)? What technical barriers exist?
>
> In its current formulation, our work allows for non-linear and learnable transformations on the already aggregated node features $\phi^{(k)}$. The authors are also interested in CLT-like results for non-linear models of aggregations. The path formalism which our current results are based on are flexible enough to handle ReLU-based gating seen in standard graph convolutional neural networks. As shown in equation (1) of [1], this would require introducing a gating random variable $H_{i,j}$ which depends on the path history of terms between nodes $i$ and $j$. This would include a path-dependency complication to the analysis however some simplifications could exist given the short-circuiting behavior of binary paths ($H_w = 0$ if $H_{w'} = 0$ and $w'$ is within $w$). A similar complication holds for the case of a learnable weight $H$ where, under the element-wise multiplication $H\circ A$, the aggregator $H\circ A$ now has paths $(H\circ A)_w$ which are not homogeneous in expectation (even in the 1-community setting). It remains an interesting open problem on how to extend our current results to the case of path-dependent aggregated quantities.
>
> > Sensitivity to $\nu_n$ Assumptions: Many results hinge on $\nu_n\to\infty$ and $\nu_n = o(n)$. Are the convergence rates or limiting distributions significantly affected if $\nu_n$ remains constant or grows slowly?
>
> The requirement that $\nu_n\to\infty$ is indeed necessary. See the empirical failure for small $\nu_n$ we discuss in response to your other question below. In the context of social graphs and other real-world networks, the setting of $\nu_n = o(n)$ is practical. Our arguments still work for the dense regime $\nu_n = \Omega(n)$ but the limiting characterization of the covariance would need to change, as the feature noise would no longer be negligible in the limit.
>
> >  Practical Implications for Model Design: Given that the class means collapse to a 1D subspace, can these results motivate architectural changes (e.g., early stopping, skip connections, regularization)?
>
> Alternatively, this may inspire additional research into flatter GNN with more expressive propagation mechanisms. Recent implementations of spectral GNNs [2,3] seem to favor this flat architecture with expressive propagators.
>
> >  Empirical Validation: Would the authors consider adding numerical simulations (beyond Figures 1–3) to illustrate how closely the empirical distributions of embeddings follow the theoretical Gaussian mixtures, especially for moderate n?
>
> Figures 4 and 5 in the supplement (Appendix G, p. 36) already provide this comparison. Even for small $n$, the empirical distribution aligns remarkably well with the predicted Gaussian mixtures. The decisive factor is the degree parameter $\nu_n$: values as modest as $\nu_n = 16$ suffice, whereas very small $\nu_n$ lead to noticeable deviations, as the plots show (and inline with our theory). In our revision, we plan to move one or two of these figures into the main paper now that we have an additional page.
>
> > Beyond CSBM: The authors mention the challenge of generalizing beyond community-based graphs. Can they speculate on potential extensions or possible failures of the current CLTs in graphs with heavy-tailed degree distributions or assortative mixing?
>
> Our framework should extend to the **degree-corrected stochastic block model (DCSBM)**, which naturally accommodates heavy-tailed degree sequences. When the (normalized) degree distribution remains stable as $n$ grows—that is, all degrees scale at the same rate—we expect the a similar CLT to hold, perhaps with some adjustments to the limit. In contrast, if the degree distribution’s tail becomes progressively heavier with $n$, the variance conditions underlying the CLTs may fail, and alternative normalizations might be required.
>
> Regarding **assortative versus disassortative mixing**, the general SBM analyzed in the paper already allows any combination of within- and between-block connection probabilities, so our results cover both assortative and disassortative regimes without modification.
>
>
> > Limitations: No — The paper does not adequately discuss limitations, although the authors indirectly allude to model constraints. I recommend: Explicitly stating the limitations of the Poly-GNN assumption (e.g., no learnable weights or non-linearities). Discussing how sensitivity to $\nu_n$ or non-community-based graphs may affect the validity of the results. Considering the gap between asymptotic and finite-n behavior, especially since real graphs are not infinitely large or cleanly community-structured.
>
> We will add an explicit **Limitations** section. Key points:
>
> - **Scope of the theory.** Our results characterize multi-hop features optimized under Lipschitz losses. They exclude nonlinear propagation and other complex GNN variants; extending to those architectures is future work.
> - **Learnable components.** Trainable functions can act on the aggregated features (the “head”), but the propagation operator itself is fixed. We will clarify this constraint.
> - **Dependence on average degree $\nu_n$.** The theory holds for $\nu_n \to \infty$ up to to $\nu_n = o(n)$; the unrealistic $\nu_n = O(1)$ regime is excluded, and $\nu_n = \Omega(n)$ requires additional arguments. Empirically, asymptotic behavior appears once $\nu_n \gtrsim 16$ (Appendix G).
> - **Model class.** We assume an SBM. While this omits truly arbitrary graphs, SBMs with large $K$ are universal $L^{2}$-graphon approximators [4–6], so the restriction is less severe than it first seems.
> - **Asymptotic vs. finite $n$.** Theorems are asymptotic; finite-sample deviations are possible. Nonetheless, experiments show excellent agreement even at modest $n$, with accuracy driven more by $\nu_n$ than by graph size.
>
>
> ---
>
> [1] Choromanska, A., Henaff, M., Mathieu, M., Ben Arous, G. & LeCun, Y.. (2015). The Loss Surfaces of Multilayer Networks.
>
> [2] Wang, X. & Zhang, M.. (2022). How Powerful are Spectral Graph Neural Networks.
>
> [3] Mingguo H., Zhewei W., Zengfeng H. & Hongteng Xu (2021). BernNet: Learning Arbitrary Graph Spectral Filters via Bernstein Approximation.
>
> [4] Wolfe, P. J., & Olhede, S. C. (2013). Non-parametric graphon estimation.
>
> [5] Orbanz, P., & Roy, D. M. (2015). Bayesian Models of Graphs, Arrays and Other Exchangeable Random Structures. IEEE Transactions on Pattern Analysis and Machine Intelligence, 37(2), 437–461.
>
> [6] Gao, C., Lu, Y., & Zhou, H. H. (2015). Rate-optimal graphon estimation. Annals of Statistics, 43(2), 762–785.

---

### Note · Authors · 2025-08-15

**Scope & contribution.** We prove CLTs for $k$-hop (monomial) GNN features on sparse community graphs (CSBM). After suitable scaling and centering, the empirical embeddings of the features converge in 1-Wasserstein to Gaussian mixtures governed by means $(\mu_\ell)$ and covariances $(\Sigma_\ell)$ depending directly on the graph community parameters $\pi$, $B$ and $M$. A central insight for the $k$-hop features is that **feature covariance collapses in the same direction as feature means**, yielding a near-1D structure as depth $k$ grows.

**What we clarified / will revise.**

• **Originality & positioning (vVTK, SM6e, BzXf).** We will expand on related work (oversmoothing, SGC/polynomial filters) and highlight our novelty: random graph CLTs for GNN features, a Wasserstein-mode of convergence which allows for limiting optimization guarantees, and the covariance-collapse mechanism (which is stronger than current methods based on power-iteration).

• **Sparse vs. dense (BzXf, SM6e).** Our theory targets the growing degree $\nu_n\to\infty$, $\nu_n=o(n)$ sparse setting where graph noise dominates. In the case of dense graphs, the feature-noise term is non-negligible and the limiting covariance changes. We will sketch these changes and fix the Assumption 1 typo (bounded $B_{\ell\ell'}$).
• **Sensitivity to $\nu_n$ (ovye).** The $\nu_n\to\infty$ condition is necessary as shown by the example of Erd\H{o}s--R\'enyi (ER) with 1 dimensional constant features. As for the sensitivity of our results, empirically the Gaussian-mixture limit is accurate already for $\nu_n\gtrsim16$.

• **Practical implications (all).** We will temper claims, clarify that degree-normalized features can use the observed average degree, and note guidance (stop or use shallow depth once collapse appears).

• **Beyond CSBM & technicals (ovye, SM6e, vVTK).** We will discuss DCSBM (heavy-tailed degrees), indicate which extensions are tedious (polynomial filters $\sum_k A^k X W_k$) versus those needing new tools (gating/attention), adjust wording, add citations, and move key Appendix-G figures showing empirical--theory fits to main text.

**Limitations (to be added).** Multi-hop non-linear feature aggregation require a separate analysis. Hurdles and promising avenues for future work will be discussed.

We will incorporate these edits in the camera-ready.

---

### Decision · Program_Chairs · 2025-09-17

**Decision:**

Accept (poster)

**Comment:**

This paper presents a novel and technically rigorous Central Limit Theorem for the embeddings of polynomial GNNs on sparse community-based graphs. The work provides a good theoretical contribution by introducing a "covariance collapse" mechanism, which shows that feature variance diminishes in the same direction as the mean signal, offering a more precise explanation for the oversmoothing phenomenon than prior work. The authors' detailed rebuttal effectively clarified the paper's originality and positioning within the literature, addressing initial reviewer concerns and leading to a clear consensus for acceptance.